

# The Met Office Unified Model Global Atmosphere 6.0/6.1 and JULES Global Land 6.0/6.1 configurations

David Walters[1], Malcolm Brooks[1], Ian Boutle[1], Thomas Melvin[1], Rachel Stratton[1], Simon Vosper[1], Helen Wells[1], Keith Williams[1], Nigel Wood[1], Thomas Allen[1], Andrew Bushell[1], Dan Copsey[1], Paul Earnshaw[1], John Edwards[1], Markus Gross[1,2], Steven Hardiman[1], Chris Harris[1], Julian Heming[1], Nicholas Klingaman[3], Richard Levine[1], James Manners[1], Gill Martin[1], Sean Milton[1], Marion Mittermaier[1], Cyril Morcrette[1], Thomas Riddick[1,4], Malcolm Roberts[1], Claudio Sanchez[1], Paul Selwood[1], Alison Stirling[1], Chris Smith[1], Dan Suri[1], Warren Tennant[1], Pier Luigi Vidale[1], Jonathan Wilkinson[1], Martin Willett[1], Steve Woolnough[3], and Prince Xavier[1]

[1]Met Office, FitzRoy Road, Exeter, EX1 3PB, UK
[2]Centro de Investigación Científica y de Educación Superior de Ensenada, Departamento de Oceanografía Física, Carretera Ensenada-Tijuana 3918, Ensenada BC 22860, Mexico
[3]National Centre for Atmospheric Science, Department of Meteorology, University of Reading, Reading, RG6 6BB, UK
[4]Max Planck Institute for Meteorology, Bundesstrasse 53, 20146 Hamburg, Germany

*Correspondence to:* David Walters (david.walters@metoffice.gov.uk)

**Abstract.** We describe Global Atmosphere 6.0 and Global Land 6.0: the latest science configurations of the Met Office Unified Model and JULES land surface model developed for use across all timescales. Global Atmosphere 6.0 includes the ENDGame dynamical core, which significantly increases mid-latitude variability improving a known model bias. Alongside developments of the model's physical parametrisations, ENDGame also increases variability in the tropics, which leads to an improved representation of tropical cyclones and other tropical phenomena. Further developments of the atmospheric and land surface parametrisations improve other aspects of model performance, including the forecasting of surface weather phenomena.

We also describe Global Atmosphere 6.1 and Global Land 6.1, which include a small number of long-standing differences from our main trunk configurations that we continue to require for operational global weather prediction.

Since July 2014, GA6.1/GL6.1 has been used by the Met Office for operational global NWP, whilst GA6.0/GL6.0 was implemented in its remaining global prediction systems over the following year.

## 1 Introduction

At the heart of all numerical models of the atmosphere is the dynamical core, which is responsible for solving the atmosphere's equations of motion. The dynamical core used by all operational configurations of the Met Office Unified Model™ (UM) prior to July 2014 was called "New Dynamics" (Davies et al., 2005). New Dynamics was introduced in 2002 and made the UM the first operational model to solve a virtually unapproximated equation set — the deep-atmosphere, non-hydrostatic equations — which was achieved using a semi-implicit semi-Lagrangian approach on a regular longitude/latitude grid. This allowed us to pursue our seamless modelling strategy and use the same dynamical core for global weather and climate predictions as for very





high resolution ($\leq 1.5$ km grid-spacing) convection permitting simulations. To solve these equations in both a stable and timely manner, however, required the application of both explicit diffusion and polar filtering and to weight the semi-implicit time stepping close to being fully implicit; this in turn numerically damped the model solution and smoothed synoptic scale features. Also, the details of how New Dynamics was applied combined with the precise layout of variables on the global grid meant

that the scalability of New Dynamics was limited to the number of computer processors typically used in operational NWP today. It has been shown not to scale over the increased number of processors that will be required in the next 5-to-10 years. For this reason, following the implementation of New Dynamics, the Met Office initiated the development of "ENDGame" (Even Newer Dynamics for General atmospheric modelling of the environment, Wood et al., 2014). ENDGame is an evolution of New Dynamics designed to maintain its benefits whilst improving its accuracy, stability and scalability. The development of

ENDGame took over 10 years and its inclusion in the Global Atmosphere 6.0 (GA6.0) configuration described herein took a further two years. The first configuration to include ENDGame was GA5.0, which combined the replacement of the dynamical core with a number of developments and improvements to the model's parametrisations. GA5.0 was frozen and assessed in 2013 but was not released for wider use. Over the following 8 months we included a number of bug-fixes, improvements and additional parametrisation developments and froze GA6.0 in October 2013. At the same time we froze a science configuration

of the JULES (Joint UK Land Environment Simulator, Best et al., 2011; Clark et al., 2011) land surface model designed for use with GA6.0: Global Land (GL6.0).

In Sect. 2 of this paper we describe GA6.0 and GL6.0, whilst in Sect. 3 we document how these differ from the last documented configurations: GA4.0 and GL4.0[1,2]. The development of these changes is documented using "trac" issue tracking software, so for consistency with that documentation, we list the trac ticket numbers along with these descriptions. For com-

pleteness, in the Appendix we also briefly outline which of these changes were included as part of GA5.0/GL5.0. In Sect. 4 we describe GA6.1 and GL6.1, which are based on the GA6.0/GL6.0 "trunk" configurations, but include a small number of long-standing changes still required for operational global NWP. In addition to outlining the motivation for these changes, we discuss our plans for removing their necessity in future releases. In July 2014, the Met Office implemented GA6.1/GL6.1 in its operational global NWP suite alongside an increase of the deterministic global model's horizontal resolution from N512

(approximately 25 km in the mid-latitudes) to N768 (approximately 17 km) and an extension of the run-length of the global ensemble from 3 to 7 days. In 2015, GA6.0/GL6.0 was implemented in the GloSea5 seasonal prediction system as part of the Global Coupled 2.0 configuration (GC2.0, documented in Williams et al., 2015) and has been used by the Met Office Hadley Centre for a series of climate change experiments as part of the HadGEM3-GC2.0 climate model.

Section 5 of the paper includes an assessment of the configuration's performance in global weather prediction and atmosphere-

only climate simulations. ENDGame's improved accuracy and reduced damping produces more detail in individual synoptic features such as cyclones, fronts, troughs and jet stream winds. In the tropics, a combination of ENDGame and improvements to the model's physics improves the UM's treatment of several modes of variability including tropical cyclones, equatorial

---

[1]Where the configurations remain unchanged from GA4.0 and GL4.0 and its predecessors, Sect. 2 contains material which is unaltered from the documentation papers for those releases (i.e. Walters et al., 2011, 2014).

[2]In addition to the material herein, the Supplement to this paper includes a short list of model settings outside the GA/GL definition that are dependent on either model resolution or system application.



Kelvin waves and the Madden–Julian Oscillation (MJO, Madden and Julian, 1971). Both ENDGame and improvements to the model's physics are shown to contribute to some significant improvements to the forecasting of near-surface weather. Finally, in Sect. 6 we outline our progress and plans for ongoing model development.

## 2   Global Atmosphere 6.0 and Global Land 6.0

### 2.1   Dynamical formulation and discretisation

The UM's ENDGame dynamical core uses a semi-implicit semi-Lagrangian formulation to solve the non-hydrostatic, fully-compressible deep-atmosphere equations of motion (Wood et al., 2014). The primary atmospheric prognostics are the three-dimensional wind components, virtual dry potential temperature, Exner pressure, and dry density, whilst moist prognostics such as the mass mixing ratio of water vapour and prognostic cloud fields as well as other atmospheric loadings are advected as free

tracers. These prognostic fields are discretised horizontally onto a regular longitude/latitude grid with Arakawa C-grid staggering (Arakawa and Lamb, 1977), whilst the vertical discretisation utilises a Charney-Phillips staggering (Charney and Phillips, 1953) using terrain-following hybrid height coordinates. The discretised equations are solved using a nested iterative approach centred about solving a linear Helmholtz equation. By convention, global configurations are defined on $2N$ longitudes and $1.5N$ latitudes of scalar grid-points with the meridional wind variable held at the north and south poles and scalar and zonal

wind variables first stored half a grid length away from the poles. This choice makes the grid-spacing approximately isotropic in the mid-latitudes and means that the integer $N$, which represents the maximum number of zonal 2 grid-point waves that can be represented by the model, uniquely defines its horizontal resolution; a model with $N = 96$ is said to be N96 resolution. Limited-area configurations use a rotated longitude/latitude grid with the pole rotated so that the grid's equator runs through the centre of the model domain. In the vertical, the majority of climate configurations use an 85 level set labelled $L85(50_t, 35_s)_{85}$,

which has 50 levels below $18\,\mathrm{km}$ (and hence at least sometimes in the troposphere), 35 levels above this (and hence solely in or above the stratosphere) and a fixed model lid $85\,\mathrm{km}$ above sea level. Limited area climate simulations use a reduced 63 level set, $L63(50_t, 13_s)_{40}$, which has the same 50 levels below $18\,\mathrm{km}$, with only 13 above and a lower model top at $40\,\mathrm{km}$. Finally, NWP configurations use a 70 level set, $L70(50_t, 20_s)_{80}$ which has an almost identical 50 levels below $18\,\mathrm{km}$, a model lid at $80\,\mathrm{km}$, but has a reduced stratospheric resolution compared to $L85(50_t, 35_s)_{85}$. Although we use a range of vertical resolutions

in the stratosphere, a consistent tropospheric vertical resolution is currently used for a given GA configuration. A more detailed description of these level sets is included in the supplementary material to this paper.

### 2.2   Structure of the atmospheric model time step

With ENDGame, the UM uses a nested iterative structure for each atmospheric time step within which processes are split into an outer loop and an inner loop. The semi-Lagrangian departure point equations are solved within the outer loop using the

latest estimates for the wind variables. Appropriate fields are then interpolated to the updated departure points. Within the inner loop, the Coriolis, orographic and non-linear terms are solved along with a linear Helmholtz problem to obtain the pressure





increment. Latest estimates for all variables are then obtained from the pressure increment via a back-substitution process; see Wood et al. (2014) for details. The physical parametrisations are split into slow processes (radiation, large-scale precipitation and gravity wave drag) and fast processes (atmospheric boundary layer, turbulence, convection and land surface coupling). The slow processes are treated in parallel and are computed once per time step before the outer loop. The source terms from the slow processes are then added on to the appropriate fields before interpolation. The fast processes are treated sequentially and are computed in the outer loop using the latest predicted estimate for the required variables at the next, $n+1$ time step. A summary of the atmospheric time step is given in Algorithm 1. In practice two iterations are used for each of the outer and inner loops so that the Helmholtz problem is solved four times per time step.

---

**Algorithm 1** Iterative structure of time step $n+1$. Here, we use two inner and two outer loops ($L=2$, $M=2$).

---

1: Given the solution at time step $n$, let the first estimate for a prognostic variable $F$ at time level $n+1$ be $F^{n+1} = F^n$

2: Compute slow parametrised processes and time level $n$ forcings $R_F^n$

3: **for** $m = 1, M$ **do** {*departure (outer-loop) iteration*}

4:     Solve the trajectory equations to compute the next estimate of the departure points using the time level $n$ and the latest estimate for time level $n+1$ wind fields

5:     Interpolate $R_F^n$ to departure points

6:     Compute time level $n+1$ predictors $F^*$

7:     Compute fast parametrised processes using latest $n+1$ predictor $F^*$

8:     Evaluate time level $n$ component of Helmholtz right hand side $\mathfrak{R}^n$

9:     **for** $l = 1, L$ **do** {*non-linear (inner-loop) iteration*}

10:         Evaluate non-linear and Coriolis terms $R_F^*$

11:         Evaluate time level $n+1$ component of Helmholtz right hand side $\mathfrak{R}^*$

12:         Solve the Helmholtz problem for the pressure increment $\pi'$ and hence obtain the next estimate for $\pi^{n+1} \equiv \pi^n + \pi'$

13:         Obtain the other prognostic variables at time level $n+1$ via back-substitution

14:     **end for**

15: **end for**

---

## 2.3 Solar and terrestrial radiation

Shortwave (SW) radiation from the Sun is absorbed in the atmosphere and at the Earth's surface and provides energy to drive the atmospheric circulation. Longwave (LW) radiation is emitted from the planet and interacts with the atmosphere, redistributing heat, before being emitted into space. These processes are parametrised via the radiation scheme, which provides prognostic atmospheric temperature increments and surface fluxes and additional diagnostic fluxes. The radiation scheme of Edwards and Slingo (1996) is used with a configuration based on Cusack et al. (1999) with a number of significant updates.





The correlated-$k$ method is used for gaseous absorption with 6 bands in the SW and 9 bands in the LW. The method of equivalent extinction (Edwards, 1996) is used for minor gases in each band. Gaseous absorption coefficients are generated using the HITRAN 2001 spectroscopic database (Rothman et al., 2003) with updates up to 2003. The water vapour continuum is represented using version 2.4 of the Clough–Kneizys–Davies (CKD) model (Clough et al., 1989; Mlawer et al., 1999). Twenty-one (21) $k$ terms are used for the major gases in the SW bands. Absorption by water vapour ($H_2O$), ozone ($O_3$), carbon dioxide ($CO_2$) and oxygen ($O_2$) is included. The treatment of $O_3$ absorption is as described in Zhong et al. (2008). The solar spectrum uses data from Lean (2000) at wavelengths shorter than $735\,nm$ with the Kurucz and Bell (1995) spectrum at longer wavelengths. Forty-seven (47) $k$ terms are used for the major gases in the LW bands. Absorption by $H_2O$, $O_3$, $CO_2$, $CH_4$, nitrous oxide ($N_2O$), CFC-11 ($CCl_3F$), CFC-12 ($CCl_2F_2$) and HFC134a ($CH_2FCF_3$) is included. For climate simulations, the atmospheric concentrations of CFC-12 and HFC134a are adjusted to represent absorption by all the remaining trace halocarbons. The treatment of $CO_2$ and $O_3$ absorption is as described in Zhong and Haigh (2000) to provide accurate stratospheric heating. Of the major gases considered, only $H_2O$ is prognostic; $O_3$ uses a zonally symmetric climatology, whilst other gases are prescribed using either fixed or time-varying mass mixing ratios and assumed to be well mixed.

Absorption and scattering by the following categories of aerosol, either prognostic or climatological, are included in both the SW and LW: ammonium sulphate, mineral dust, sea salt, biomass burning, fossil-fuel black carbon, fossil-fuel organic carbon, and secondary organic (biogenic) aerosols. The parametrisation of cloud droplets is described in Edwards and Slingo (1996) using the method of "thick averaging". Padé fits are used for the variation with effective radius, which is computed from the number of cloud droplets. This cloud droplet number is derived from either prognostic or climatological aerosol concentrations in all modelling systems (Jones et al., 1994, 2001). The parametrisation of ice crystals is described in Edwards et al. (2007). Full treatment of scattering is used in both the SW and LW. The sub-grid cloud structure is represented using the Monte Carlo Independent Column Approximation (McICA) as described in Hill et al. (2011), with optimal sampling using 6 extra terms in the LW and 10 in the SW for the reduction of random noise.

Full radiation calculations are made every hour using the instantaneous cloud fields and a mean solar zenith angle for the following 1 h period. Corrections are made for the change in solar zenith angle on every model time step as described in Manners et al. (2009). The emissivity and the albedo of the surface are set by the land surface model. The direct SW flux at the surface is corrected for the angle and aspect of the topographic slope as described in Manners et al. (2012). The albedo of the sea surface uses a modified version of the parametrisation from Barker and Li (1995) with a varying spectral dependence.

### 2.4 Large-scale precipitation

The formation and evolution of precipitation due to grid scale processes is the responsibility of the large-scale precipitation — or microphysics — scheme, whilst small-scale precipitating events are handled by the convection scheme. The microphysics scheme has prognostic input fields of temperature, moisture and cloud from the end of the previous time step, which it modifies in turn. The microphysics used is based on Wilson and Ballard (1999), with extensive modifications. We use a prognostic rain formulation, which allows three-dimensional advection of the precipitation mass mixing ratio. The particle size distribution for rain uses rain-rate dependent distribution of Abel and Boutle (2012). The minimum cloud liquid content for autoconversion to





occur has been altered from the original Tripoli and Cotton (1980) formulation to a liquid content where the number of drops over $20\,\mu m$ is $1000\,cm^{-3}$, as discussed in Abel et al. (2010). In addition, we have used the fall velocities of Abel and Shipway (2007), which allow a better representation of the sedimentation of small droplets. We also make use of multiple sub-time steps of the precipitation scheme, as in Posselt and Lohmann (2008) with one sub-time step for every two minutes of the model time

step to achieve a realistic treatment of in-column evaporation. Aerosol mass mixing ratios provide the cloud droplet number for autoconversion, according to the formulae of Jones et al. (1994, 2001). The aerosols which provide the droplet number are ammonium sulphate, sea salt, biomass burning and fossil-fuel organic carbon. When using climatological aerosol, the cloud droplet number is the same as that used in the radiation scheme.

## 2.5 Large-scale cloud

Clouds appear on sub-grid scales well before the humidity averaged over the size of a model grid box reaches saturation. A cloud parametrisation scheme is therefore required to determine the fraction of the grid box which is covered by cloud and the amount and phase of condensed water contained in those clouds. The formation of clouds will convert water vapour into liquid or ice and release latent heat. The cloud cover and liquid and ice water contents are then used by the radiation scheme to calculate the radiative impact of the clouds and by the large-scale precipitation scheme to calculate whether any precipitation

has formed.

 The parametrisation used is the prognostic cloud fraction and prognostic condensate (PC2) scheme (Wilson et al., 2008a, b) along with the modifications to the cloud erosion parametrisation described by Morcrette (2012). PC2 uses three prognostic variables for water mixing ratio — vapour, liquid and ice — and a further three prognostic variables for cloud fraction: liquid, ice and mixed-phase. The following atmospheric processes can modify the cloud fields: shortwave radiation, longwave

radiation, boundary layer processes, convection, precipitation, small-scale mixing (cloud erosion), advection and changes in atmospheric pressure. The convection scheme calculates increments to the prognostic liquid and ice water contents by detraining condensate from the convective plume, whilst the cloud fractions are updated using the non-uniform forcing method of Bushell et al. (2003). One advantage of the prognostic approach is that clouds can be transported away from where they were created. For example, anvils detrained from convection can persist and be advected downstream long after the convection itself

has ceased.

## 2.6 Sub-grid orographic drag

The effect of local and mesoscale orographic features not resolved by the mean orography, from individual hills through to small mountain ranges, must be parametrised. The smallest scales, where buoyancy effects are not important, are represented by an effective roughness parametrisation in which the roughness length for momentum is increased above the surface roughness

to account for the additional stress due to the sub-grid orography (Wood and Mason, 1993). The effects of the remainder of the sub-grid orography (on scales where buoyancy effects are important) are parametrised by a drag scheme which represents the effects of low-level flow blocking and the drag associated with stationary gravity waves (mountain waves). This is based on the scheme described by Lott and Miller (1997), but with some important differences, described in more detail in Sect. 3.5.





The sub-grid orography is assumed to consist of uniformly distributed elliptical mountains within the grid box, described in terms of a height amplitude, which is proportional to the grid-box standard deviation of the source orography data, anisotropy (the extent to which the sub-grid orography is ridge-like, as opposed to circular), the alignment of the major axis and the mean slope along the major axis. The scheme is based on two different frameworks for the drag mechanisms: bluff body dynamics for the flow-blocking and linear gravity waves for the mountain wave drag component.

The degree to which the flow is blocked and so passes around, rather than over the mountains is determined by the Froude number, $F = U/(NH)$ where $H$ is the assumed sub-grid mountain height (proportional to the sub-grid standard deviation of the source orography data) and $N$ and $U$ are respectively measures of the buoyancy frequency and wind speed of the low-level flow. When $F$ is less than the critical value, $F_c$, a fraction of the flow is assumed to pass around the sides of the orography, and a drag is applied to the flow within this blocked layer. Mountain waves are generated by the remaining proportion of the layer, through which the orography pierces through. The acceleration of the flow due to wave stress divergence is exerted at levels where wave breaking is diagnosed.

## 2.7 Non-orographic gravity wave drag

Non-orographic sources — such as convection, fronts and jets — can force gravity waves with non-zero phase speed. These waves break in the upper stratosphere and mesosphere, depositing momentum, which contributes to driving the zonal mean wind and temperature structures away from radiative equilibrium. Waves on scales too small for the model to sustain explicitly are represented by a spectral sub-grid parametrisation scheme (Scaife et al., 2002), which by contributing to the deposited momentum leads to a more realistic tropical quasi-biennial oscillation. The scheme, described in more detail in Walters et al. (2011), represents processes of wave generation, conservative propagation and dissipation by critical-level filtering and wave saturation acting on a vertical wavenumber spectrum of gravity wave fluxes following Warner and McIntyre (2001). Launched in the lower troposphere, the two-part spectrum is linear from low wavenumber cut-off up to a spectrum peak, corresponding to wavelengths of 20 km and 4.3 km, whilst beyond the peak an inverse cubic tail is characteristic of saturation. Current values chosen to scale the spectrum represent of order 10 % of the saturation spectrum amplitudes at launch height. Momentum conservation is enforced at launch, where isotropic fluxes guarantee zero net momentum, and by imposing a condition of zero vertical wave flux at the model upper boundary. In between, momentum deposition occurs in each layer where reduced integrated flux results from erosion of the launch spectrum, after transformation by conservative propagation, to match the locally evaluated saturation spectrum.

## 2.8 Atmospheric boundary layer

Turbulent motions in the atmosphere are not resolved by global atmospheric models, but are important to parametrise in order to give realistic vertical structure in the thermodynamic and wind profiles. Although referred to as the "boundary layer" scheme, this parametrisation represents mixing over the full depth of the troposphere. The scheme is that of Lock et al. (2000) with the modifications described in Lock (2001) and Brown et al. (2008). It is a first-order turbulence closure mixing adiabatically conserved heat and moisture variables, momentum and tracers. For unstable boundary layers, diffusion coefficients



($K$ profiles) are specified functions of height within the boundary layer, related to the strength of the turbulence forcing. Two separate $K$ profiles are used, one for surface sources of turbulence (surface heating and wind shear) and one for cloud-top sources (radiative and evaporative cooling). The existence and depth of unstable layers is diagnosed initially by moist adiabatic parcels and then adjusted to ensure that the buoyancy consumption of turbulence kinetic energy is limited. This can permit

the cloud layer to decouple from the surface (Nicholls, 1984). If cumulus convection is diagnosed (through comparison of cloud and sub-cloud layer moisture gradients), the surface-driven $K$ profile is restricted to below cloud base and the mass flux convection scheme is triggered from that level. Mixing across the top of the boundary layer is through an explicit entrainment parametrisation that is coupled to the radiative fluxes and the dynamics through a sub-grid inversion diagnosis. If the thermo-dynamic conditions are right, cumulus penetration into a stratocumulus layer can generate additional turbulence and cloud-top

entrainment in the stratocumulus by enhancing evaporative cooling at cloud top. There are additional non-local fluxes of heat and momentum in order to generate more vertically uniform potential temperature and wind profiles in convective boundary layers. For stable boundary layers and in the free troposphere, we use a local Richardson number scheme based on Smith (1990). Its stable stability dependence is given by the "sharp" function over sea and by the "MES-tail" function over land (which matches linearly between an enhanced mixing function at the surface and "sharp" at $200\,\mathrm{m}$ and above). This additional

near-surface mixing is motivated by the effects of surface heterogeneity, such as those described in McCabe and Brown (2007). The resulting diffusion equation is solved implicitly using the monotonically damping, second-order-accurate, unconditionally stable numerical scheme of Wood et al. (2007). The kinetic energy dissipated through the turbulent shear stresses is returned to the atmosphere as a local heating term.

## 2.9 Convection

The convection scheme represents the sub-grid scale transport of heat, moisture and momentum associated with cumulus clouds within a grid box. The UM uses a mass flux convection scheme based on Gregory and Rowntree (1990) with various extensions to include down-draughts (Gregory and Allen, 1991) and convective momentum transport (CMT). The current scheme consists of three stages: (i) convective diagnosis to determine whether convection is possible from the boundary layer; (ii) a call to the shallow or deep convection scheme for all points diagnosed deep or shallow by the first step; and (iii) a call to the mid-level

convection scheme for all grid points.

The diagnosis of shallow and deep convection is based on an undilute parcel ascent from the near surface for grid boxes where the surface layer is unstable and forms part of the boundary layer diagnosis (Lock et al., 2000). Shallow convection is then diagnosed if the following conditions are met: (i) the parcel attains neutral buoyancy below $2.5\,\mathrm{km}$ or below the freezing level, whichever is higher, and (ii) the air in model levels forming a layer of order $1500\,\mathrm{m}$ above this has a mean upward vertical

velocity less than $0.02\,\mathrm{m\,s^{-1}}$. Otherwise, convection diagnosed from the boundary layer is defined as deep.

The deep convection scheme differs from the original Gregory and Rowntree (1990) scheme in using a convective available potential energy (CAPE) closure based on Fritsch and Chappell (1980). Mixing detrainment rates now depend on relative humidity and forced detrainment rates adapt to the buoyancy of the convective plume (Derbyshire et al., 2011). The CMT scheme uses a flux gradient approach (Stratton et al., 2009).





The shallow convection scheme uses a closure based on Grant (2001) and has larger entrainment rates than the deep scheme consistent with cloud-resolving model (CRM) simulations of shallow convection. The shallow CMT uses flux–gradient relationships derived from CRM simulations of shallow convection (Grant and Brown, 1999).

The mid-level scheme operates on any instabilities found in a column above the top of deep or shallow convection or above the lifting condensation level. The scheme is largely unchanged from Gregory and Rowntree (1990), but uses the Gregory et al. (1997) CMT scheme and a CAPE closure. The mid-level scheme operates mainly either overnight over land when convection from the stable boundary layer is no longer possible or in the region of mid-latitude storms. Other cases of mid-level convection tend to remove instabilities over a few levels and do not produce much precipitation.

The timescale for the CAPE closure, which is used for the deep and mid-level convection schemes, is essentially fixed at a chosen value of one hour; however, if extremely high large-scale vertical velocities are detected in the column then the timescale is rapidly reduced to ensure numerical stability.

### 2.10 Atmospheric aerosols and chemistry

As discussed in Walters et al. (2011), the modelling of atmospheric aerosols and chemistry is considered as a separate component of the full Earth system and remains outside the scope of this document. The aerosol species represented and their interaction with the atmospheric parametrisations is, however, part of the Global Atmosphere component and has therefore been included in the descriptions above. Systems including prognostic aerosol modelling do so using the CLASSIC (Coupled Large-scale Aerosol Simulator for Studies in Climate) aerosol scheme described in Bellouin et al. (2011), whilst systems not including prognostic aerosols use a three-dimensional monthly climatology for each aerosol species to model both the direct and indirect aerosol effects. In addition to the treatment of these tropospheric aerosols, we include a simple stratospheric aerosol climatology based on Cusack et al. (1998). We also include the production of stratospheric water vapour via a simple methane oxidation parametrisation (Untch and Simmons, 1999).

### 2.11 Land surface and hydrology: Global Land 6.0

The exchange of fluxes between the land surface and the atmosphere is an important mechanism for heating and moistening the atmospheric boundary layer. In addition, the exchange of $CO_2$ and other greenhouse gases plays a significant role in the climate system. The hydrological state of the land surface contributes to impacts such as flooding and drought as well as providing freshwater fluxes to the ocean, which influences ocean circulation. Therefore, a land surface model needs to be able to represent this wide range of processes over all surface types that are present on the Earth.

The Global Land configuration uses a community land surface model, JULES (Best et al., 2011; Clark et al., 2011), to model all of the processes at the land surface and in the sub-surface soil. A tile approach is used to represent sub-grid scale heterogeneity (Essery et al., 2003), with the surface of each land point subdivided into five types of vegetation (broadleaf trees, needle-leaved trees, temperate C3 grass, tropical C4 grass and shrubs) and four non-vegetated surface types (urban areas, inland water, bare soil and land ice). Vegetation canopies are represented in the surface energy balance through the coupling to the underlying soil. This canopy is coupled via radiative and turbulent exchange, whilst bare soil beneath the



canopy component is coupled through conduction. JULES also uses a canopy radiation scheme to represent the penetration of light within the vegetation canopy and its subsequent impact on photosynthesis (Mercado et al., 2007). The canopy also interacts with the surface snow. For most vegetation types, the snow is held on top of the canopy, whilst for needle-leaved trees the interception of snow by the canopy is represented with separate snow stores on the canopy and on the ground. This impacts the surface albedo, the snow sublimation and the snow melt. The vegetation canopy code has been adapted for use with the urban surface type by defining an "urban canopy" with the thermal properties of concrete (Best, 2005). This has been demonstrated to give improvements over representing an urban area as a rough bare soil surface. Similarly, this canopy approach has also been adopted for the representation of lakes. The original representation was through a soil surface that could evaporate at the potential rate (i.e. a soggy soil), which has been shown to have incorrect seasonal and diurnal cycles for the surface temperature (Rooney and Jones, 2010). By defining an "inland water canopy" and setting the thermal characteristics to those of a suitable depth of water (taken to be $1\,\mathrm{m}$), a better diurnal cycle for the surface temperature is achieved.

Surface fluxes are calculated separately on each tile using surface similarity theory. In stable conditions we use the similarity functions of Beljaars and Holtslag (1991), whilst in unstable conditions we take the functions from Dyer and Hicks (1970). The effects on surface exchange of both boundary layer gustiness (Godfrey and Beljaars, 1991) and deep convective gustiness (Redelsperger et al., 2000) are included. Temperatures at $1.5\,\mathrm{m}$ and winds at $10\,\mathrm{m}$ are interpolated between the model's grid levels using the same similarity functions, but a parametrisation of transitional decoupling in very light winds is included in the calculation of the $1.5\,\mathrm{m}$ temperature.

Soil processes are represented using a 4-layer scheme for the heat and water fluxes with hydraulic relationships taken from van Genuchten (1980). These four soil layers have thicknesses from the top down of 0.1, 0.25, 0.65 and $2.0\,\mathrm{m}$. The impact of moisture on the thermal characteristics of the soil is represented using a simplification of Johansen (1975), as described in Dharssi et al. (2009). The energetics of water movement within the soil is accounted for, as is the latent heat exchange resulting from the phase change of soil water from liquid to solid states. Sub-grid scale heterogeneity of soil moisture is represented using the Large-Scale Hydrology approach (Gedney and Cox, 2003), which is based on the topography-based rainfall-runoff model TOPMODEL (Beven and Kirkby, 1979). This enables the representation of an interactive water table within the soil that can be used to represent wetland areas, as well as increasing surface runoff through heterogeneity in soil moisture driven by topography.

A river routing scheme is used to route the total runoff from inland grid points both out to the sea and to inland basins, where it can flow back into the soil moisture. Excess water in inland basins is distributed evenly across all sea outflow points. In coupled model simulations the resulting freshwater outflow is passed to the ocean, where it is an important component of the thermohaline circulation, whilst in atmosphere/land-only simulations this ocean outflow is purely diagnostic. River routing calculations are performed using the TRIP (Total Runoff Integrating Pathways) model (Oki and Sud, 1998), which uses a simple advection method (Oki, 1997) to route total runoff along prescribed river channels on a $1° \times 1°$ grid using a $3\,\mathrm{h}$ time step. Land surface runoff accumulated over this time step is mapped onto the river routing grid prior to the TRIP calculations, after which soil moisture increments and total outflow at river mouths are mapped back to the atmospheric



grid (Falloon and Betts, 2006). This river routing model is not currently being used in limited-area or NWP implementations of the Global Atmosphere/Land.

### 2.12 Ancillary files and forcing data

In the UM, the characteristics of the lower boundary, the values of climatological fields and the distribution of natural and
5 anthropogenic emissions are specified using ancillary files. Use of correct ancillary file inputs can play as important a role in the performance of a system as the correct choice of many options in the parametrisations described above. For this reason, we consider the source data and processing required to create ancillaries as part of the definition of the Global Atmosphere/Land configurations. Table 1 contains the main ancillaries used as well as references to the source data from which they are created.

| Ancillary field | Source data | Notes |
|---|---|---|
| Land mask/fraction | System dependent | |
| Mean/sub-grid orography | GLOBE $30''$; Hastings et al. (1999) | Fields filtered before use |
| Land usage | IGBP; Global Soil Data Task (2000) | Mapped to 9 tile types |
| Soil properties | HWSD; Nachtergaele et al. (2008) | |
| Leaf area index/canopy height | MODIS collection 5 | 4 km data (Samanta et al., 2012) mapped to 5 plant types |
| Bare soil albedo | MODIS; Houldcroft et al. (2008) | |
| Snow free surface albedo | GlobAlbedo; Muller et al. (2012) | Spatially complete white sky values |
| TOPMODEL topographic index | Verdin and Jensen (1996) | |
| SST/sea ice | System/experiment dependent | |
| Ozone | SPARC-II; Cionni et al. (2011) | Zonal mean field used[%] |
| Aerosol emissions/fields: | | Only required for prognostic aerosol simulations |
|     Main primary emissions | CMIP5; Lamarque et al. (2010) | Includes $SO_2$, DMS, soot, OCFF, biomass burning |
|     Volcanic $SO_2$ emissions | Andres and Kasgnoc (1998) | |
|     Sulphur-cycle offline oxidants | STOCHEM[*] Derwent et al. (2003) | |
|     Ocean DMS concentrations | Kettle et al. (1999) | |
|     Biogenic aerosol ancillary | STOCHEM[*]; Derwent et al. (2003) | |
| CLASSIC aerosol climatologies | System/experiment dependent | Used when prognostic fields not available |
| TRIP river paths | $1°$ data from Oki and Sud (1998) | Adjusted at coastlines to ensure correct outflow |

**Table 1.** Source datasets used to create standard ancillary files used in GA6.0/GL6.0. [*]STOCHEM denotes that these fields are derived from runs of the STOCHEM chemistry model. [%]This is expanded to a "zonally symetric" 3D field in limited area simulations on a rotated pole grid.



## 3  Developments since Global Atmosphere/Land 4.0

The previous section provides a general description of the whole of the GA6.0 and GL6.0 configurations. In this section, we describe in more detail how these configurations differ from the previously documented configurations of GA4.0 and GL4.0.

### 3.1  Dynamical formulation and discretisation

**Introduction of the ENDGame dynamical core (GA tickets #18, 93, 94, 106 and 126[3])**

By far the largest change in GA6.0 is that we replace the "New Dynamics" dynamical core with "ENDGame" (Even Newer Dynamics for General Atmospheric Modelling of the Environment Wood et al., 2014). ENDGame and New Dynamics share many aspects of their design; both employ semi-implicit semi-Lagrangian finite-difference discretisations of the deep-atmosphere non-hydrostatic equations and both are discretised on a latitude-longitude grid with a C-grid/Charney-Phillips staggering in the
horizontal/vertical.

There are, however, a number of areas in which ENDGame differs from New Dynamics. The overall motivation behind updating the dynamical core is to retain the beneficial aspects of New Dynamics, but to improve a number of areas where it was found to be deficient, with the principal aims of improving the accuracy, scalability and stability of the model. Here, we list the most significant differences between ENDGame and New Dynamics, all of which are designed to impact at least one
of these areas.

– The nested iterative time stepping structure provides better numerical stability and allows the temporal off-centring of the trapezoidal scheme to be reduced. Time averaged terms are split as $\overline{F}^{t} = \alpha F^{n+1} + (1 - \alpha) F_{D}^{n}$. In New Dynamics, the off-centring parameter, $\alpha$, typically takes a value $0.7$ or $1$ whilst in ENDGame this is reduced to $0.55$. This has the effect of improving the model's accuracy (it would be second order accurate for $\alpha = 0.5$).

– The trajectory equation uses a centred iterative approximation: $u^{n+1/2} = 1/2 \left( u^{n+1} + u_{D}^{n} \right)$ for the velocities at the midpoint of a trajectory, replacing the extrapolated estimate: $u^{n+1/2} = 3/2 u^{n} - 1/2 u^{n-1}$, improving the stability of the model.

– The iterative solver allows more terms, such as Coriolis and orographic terms (which were previously handled explicitly or in the Helmholtz equation), to be treated as part of the nested iteration procedure and therefore a simplified Helmholtz
equation is formed, which improves scalability on parallel machines. This partly mitigates the increased cost of solving the Helmholtz equation multiple times per time step due to the nested approach.

– Virtual dry potential temperature, $\theta_{vd} \equiv \theta \left( 1 + m_{v}/\epsilon \right)$, is used as the prognostic thermodynamic variable. In addition, the non-interpolating in the vertical advection scheme, that was used for potential temperature, is replaced by a full three-dimensional semi-Lagrangian interpolation to be consistent with other fields (see Wood et al. (2014) for details).

---

[3]See the appendix for details of these individual "tickets".




- The continuity equation is discretised in a semi-Lagrangian manner instead of Eulerian to be fully consistent with the other discretised equations. This improves the accuracy and stability of the model, particularly in polar regions where the semi-implicit Eulerian discretisation of the New dynamics unphysically slows down advection compared with the semi-Lagrangian method used for other variables. This comes at the cost of losing inherent mass conservation unless a computationally expensive conservative semi-Lagrangian scheme is used, such as that used in Wood et al. (2014). Here, however, for computational efficiency, we employ a simple mass fixer to regain mass conservation.

- The ENDGame horizontal grid is shifted half a grid length in both the zonal and meridional directions compared to New Dynamics. Therefore, scalars are no longer held at the grid singularity and hence no Helmholtz equation is solved at the poles. Moreover, far fewer communications are required between polar processors to maintain the consistency of scalar fields at the pole.

- No polar filter is used in ENDGame. Since the polar filter requires multiple sweeps along near-polar rows, and hence communication across polar processors, this change further improves the scalability of the model. Furthermore, the targeted diffusion of moisture in areas with strong updrafts, designed to improve the stability of the model, is no longer used in GA6.0.

- As described in Sect. 2.2 the fast parametrised processes are now handled in the outer iterative loop. This provides a tighter coupling between the resolved dynamics and parameterisations allowing a better estimate of the time-level $n+1$ fields to be used for the parameterisations, but at the cost that they are now called once for each outer loop iteration, instead of just once per time step.

- Moist prognostics are handled in terms of mass mixing ratios instead of the specific quantities used in New Dynamics. Where they are needed for the physical parametrisations, specific quantities are converted from the mixing ratios as part of the time step.

The interested reader is directed to Wood et al. (2014) and Davies et al. (2005) for further detail.

The improved numerical accuracy and stability of the model allows it to run without the polar filter, explicit horizontal diffusion and targeted diffusion and allows the semi-implicit off-centring weights ($\alpha$) being much closer to a centred scheme. The latter of these changes leads to a reduction in implicit damping of the solution, which is the largest improvement in the physical accuracy of the model; this is discussed in more detail in Sect. 5. The improved scaling performance of ENDGame on large processor counts can be seen in Fig. 1. ENDGame continues to show scalability out to $\sim 7000$ computational cores whilst New Dynamics does not scale beyond $\sim 4000$ cores.

In addition to the changes mentioned above, there are a number of differences between the setup of the dynamical core for GA6.0 and that used for the idealised tests presented in Wood et al. (2014). Most of these are intended to improve the computational performance of the model.

- All the semi-implicit off-centring and relaxation parameters ($\tau$) are set to $\alpha = \tau = 0.55$ instead of $0.5$.





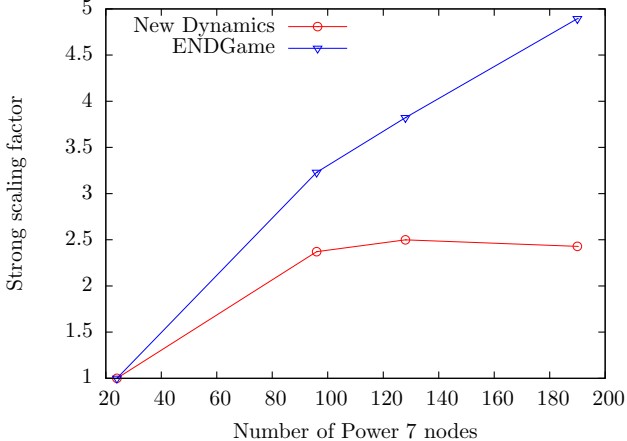

**Figure 1.** Strong scaling plot simulations using the ENDGame and New Dynamics dynamical cores (both using GA3.1 physics and full operational diagnostic and input/output loads) at N768 resolution on an IBM Power 7 supercomputer. Each Power 7 node contains 32 compute cores.

– The non-conserving version of the continuity equation is used. To obtain the density, as part of the back-substitution process, the equation of state is used instead of the continuity equation. With these changes it was found that the model could be run stably with a larger tolerance to the Helmholtz solver, thus increasing the computational performance of the model. To obtain mass conservation, an a-posteriori mass fixer is applied to the density at the end of each timestep to ensure the total mass of the atmosphere is conserved without altering the potential energy. This involves multiplying the density field, $\rho^{n+1}$, by a height dependent function to ensure mass is conserved, i.e.

$$\rho^{n+1} \longrightarrow (A + Bz)\rho^{n+1}, \tag{1}$$

where $A$ and $B$ are computed so that the total mass $M^0$ and the current estimate for the potential energy $P^{n+1}$ are unchanged

$$\sum_{i,j,k} (A + Bz_{ijk})\rho_{ijk}^{n+1}V_{ijk} = M^0, \tag{2}$$

$$\sum_{i,j,k} (A + Bz_{ijk})gz_{ijk}\rho_{ijk}^{n+1}V_{ijk} = P^{n+1}, \tag{3}$$

where $V_{ijk}$ is the volume of grid-cell $i, j, k$. Additionally, we approximately preserve the current estimate of the internal energy

$$I^{n+1} \equiv \sum_{i,j,k} c_{vd} \left( \overline{\theta_{vd}^{n+1}}^k \rho_d^{n+1} \right)_{ijk} V_{ijk}, \tag{4}$$

where $c_{vd}$ is the heat capacity of dry air at constant volume, by modifying $\theta_{vd}^{n+1}$ inversely

$$\theta_{vd}^{n+1} \longrightarrow \theta_{vd}^{n+1}/(A + Bz) \tag{5}$$

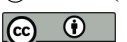


which for $B = 0$ will preserve the internal energy but due to the averaging of $\theta_{vd}$ in (5) will only be approximate when $B \neq 0$.

– Although run in the deep-atmosphere, non-hydrosatic mode, the GA6.0 implementation of ENDGame uses constant gravity, i.e. the $(a/r)^2$ variation is neglected, as would be done in a shallow-atmosphere approximation.

– Although, as mentioned previously, polar filtering is not used, to control noise in the polar regions the implicit damping layer on the vertical velocity described in Wood et al. (2014) is extended to cover all heights in the polar regions, i.e. the definition of $\mu$ in Wood et al. (2014), their (77), becomes

$$\mu(\phi,\eta) = \begin{cases} 0 & 0 \leq \eta^* < \eta_B \\ \bar{\mu} \left\{ \begin{array}{c} \sin^2\left[\frac{\pi}{2}\left(\frac{\eta^*-\eta_B}{1-\eta_B}\right)\right] \\ +\sin^{40}(\phi) \end{array} \right\} & \eta_B \leq \eta^* \leq 1 \end{cases} \tag{6}$$

with $\eta^* = 1 + (\eta - 1)\cos(\phi)$, Fig. 2 shows the geographical extent of the sponge layer for $\eta_B = 1/2$.

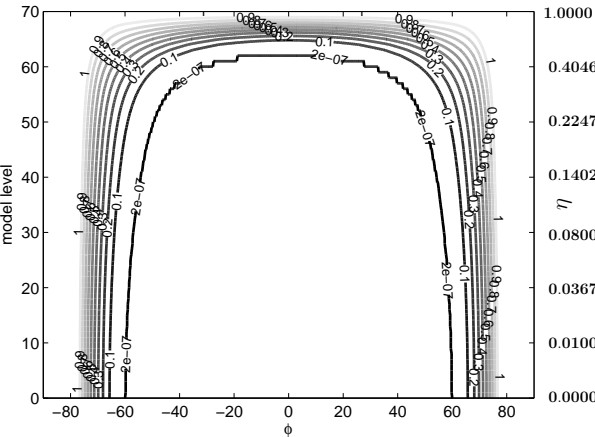

**Figure 2.** Implicit sponge layer $\mu(\phi,\eta)$ as given by (6) with $\eta_B = 1/2$ and $\bar{\mu} = 1$. Left axis shows model levels, using L70(50t, 20s)$_{80}$, right axis shows computational $\eta$ space (physical height $z \sim \eta$).

– Instead of the iterated trapezoidal method of solving the departure point equations as outlined in Wood et al. (2010) a simpler total variance diminishing $2^{nd}$-order Runge-Kutta method (TVD-RK2, see for example Shu and Osher, 1988) is used. This change is intended to reduce the number of interpolations performed in each departure point computation, and therefore reduces the amount of communication needed between processors when the departure points lie off-processor, e.g. in areas with a large horizontal Courant number. In the first outer loop a single Euler step is used to get an estimate of the departure point instead of two iterations of the trapezoidal method. In the second outer loop, both stages of the TVD-RK2 scheme are used to obtain the departure point. For a prototypical departure point equation

$$\mathbf{x}_t = \mathbf{u}, \tag{7}$$





the two schemes can be compared as shown in Table 2. Only the second stage of the TVD-RK2 method involves interpolation to a departure point, compared with at every iteration of the trapezoidal method. Thus, the total number of interpolations (which contribute the major computational and communication cost of the scheme) in computing the departure points is reduced by a factor of 4.

| (Outer iteration,Stage) | Trapezoidal | TVD-RK2 |
|---|---|---|
| $(1,1)$ | $\mathbf{x}_D^{(1,1)} = \mathbf{x}_A - \frac{\Delta t}{2}\left[\mathbf{u}^n\left(\mathbf{x}_D^{(0)}\right) + \mathbf{u}^n\right]$ | $\mathbf{x}_D^{(1,1)} = \mathbf{x}_A - \Delta t\mathbf{u}^n$ |
| $(1,2)$ | $\mathbf{x}_D^{(1,2)} = \mathbf{x}_A - \frac{\Delta t}{2}\left[\mathbf{u}^n\left(\mathbf{x}_D^{(1,1)}\right) + \mathbf{u}^{n+1}\right]$ | $\mathbf{x}_D^{(1,2)} \equiv \mathbf{x}_D^{(1,1)}$ |
| $(2,1)$ | $\mathbf{x}_D^{(2,1)} = \mathbf{x}_A - \frac{\Delta t}{2}\left[\mathbf{u}^n\left(\mathbf{x}_D^{(1,2)}\right) + \mathbf{u}^{n+1}\right]$ | $\mathbf{x}_D^{(2,1)} = \mathbf{x}_A - \Delta t\mathbf{u}^{n+1}$ |
| $(2,2)$ | $\mathbf{x}_D^{(2,2)} = \mathbf{x}_A - \frac{\Delta t}{2}\left[\mathbf{u}^n\left(\mathbf{x}_D^{(2,1)}\right) + \mathbf{u}^{n+1}\right]$ | $\mathbf{x}_D^{(2,2)} = \mathbf{x}_A - \frac{\Delta t}{2}\left[\mathbf{u}^n\left(\mathbf{x}_D^{(2,1)}\right) + \mathbf{u}^{n+1}\right]$ |

**Table 2.** Comparison of the departure point calculations as described in Wood et al. (2014) (Trapezoidal) and as used in GA6.0 (TVD-RK2). Subscripts $A$ and $D$ denote the arrival (grid) and departure points respectively.

 – The three-dimensional semi-Lagrangian interpolation of virtual dry potential temperature uses cubic Hermite rather than cubic Lagrange vertical interpolation. This removes a spurious numerical source of heating at the tropopause, where there are small vertical oscillations (Hardiman et al., 2015).

 – For most simulations, we run a fully implicit first timestep ($\alpha = 1$) to remove any spurious motion due to a lack of quasi-hydrostatic balance from either changing the dynamical core, regridding the initial state from another model/resolution or introducing analysis increments for data assimilation.

## 3.2 Structure of the atmospheric model time step

**Improvement to the conservation of water (GA tickets #75 and #78)**

As discussed in Sect. 4.5.1 of Walters et al. (2014), in GA4.0/GL4.0 the imbalance between the global mean precipitation and the global mean evaporation — denoted "$P-E$" — was deemed large enough that the configuration could not be used for long coupled climate simulations. This was found to be largely due to errors in the conversions between mixing ratios and specific quantities when using mixing ratios for the moist prognostics in the slow physics schemes. As discussed in Sect. 3.1, however, this does require multiple conversions of the moist prognostics per time step. For GA6.0, we negate this error by temporarily reverting to using specific quantities in all the physical parametrisations. We further improve the imbalance by consistently using volume averaging when interpolating between different coordinate types on the vertically-staggered grid, rather than using a mixture of volume averaging and linear interpolation. In a 50 year integration of the coupled climate model at N96 resolution ($\approx 135\,\mathrm{km}$ in the mid-latitudes), this reduces the global $P-E$ from $4\times 10^{-3}\,\mathrm{mm\,day^{-1}}$ to $-1\times 10^{-5}\,\mathrm{mm\,day^{-1}}$, which is deemed acceptable for use in long coupled integrations.





## 3.3 Solar and terrestrial radiation

**Reduced radiation time step (GA ticket #70)**

At GA4.0, full radiation calculations were made every 3 hours, with corrections for the change in cloud fields made every hour as described in Manners et al. (2009). This is replaced in GA6.0 with full radiation calculations every hour. This provides an
improved treatment of the diurnal cycle due to the proper treatment of solar zenith angle, temperature, aerosol and water vapour changes each hour. The treatment of cloud within radiation is correspondingly more consistent with McICA sampling of sub-grid cloud, now being done every hour, reducing the effects of sampling noise. Full hourly radiation also provides the potential for an improved frequency of diagnostic output from all models and of coupling with the ocean in coupled configurations. The CPU time spent within the radiation code is roughly doubled with this change. The fractional increase in the full atmospheric
model runtime is dependent on the system in which it is applied, but is of the order of 5% for global NWP applications.

## 3.4 Large-scale cloud

**Changes to the treatment of mixed-phase cloud (GA ticket #43)**

At GA6.0, we use the mixed-phase cloud fraction as PC2's third prognostic cloud fraction variable, which is a change from the original Wilson et al. (2008a) formulation used until GA4.0. It was found that the numerics of advecting this quantity were
better than the previously used total cloud fraction; this ensures that the total cloud fraction is always consistent with its three constituent parts, which was not previously the case.

## 3.5 Orographic drag

### 3.5.1 Introduction of the 5A gravity wave drag scheme (GA ticket #10)

Major changes to parametrisations in the UM are indicated by incrementing the "version" of the scheme, with each version
denoted by a number/letter combination. Previous Global Atmosphere configurations used the 4A orographic gravity wave drag scheme, described by Webster et al. (2003); GA6.0 uses the new 5A orographic drag scheme, described in detail by Wells (2015) and Vosper (2015). The description below is taken largely from the latter publication.

    The 4A scheme used a single expression for the total surface stress, which is partitioned into mountain wave and flow blocking components due to flow over, and around the orography, respectively. The new 5A scheme is based on two separate
conceptual models: bluff body dynamics for the flow-blocking drag and linear gravity-wave theory for the mountain wave drag. This approach allows for greater flexibility since the two drag mechanisms can be treated more independently.

    In more detail, the 5A scheme closely follows that described by Lott and Miller (1997), but with the following modifications:

– The original Lott and Miller (1997) scheme is modified to represent a "cut-off mountain" where only the proportion of the orography above the blocked flow layer contributes to the mountain-wave drag. This approach is also used in the
ECMWF implementation of the scheme.





- Based on the study by Vosper et al. (2009), an alternative averaging depth is used to calculate the Froude number and the depth of blocked flow layer.

- Where wave breaking is diagnosed, the wave drag is applied over an estimate for the vertical wavelength of a hydrostatic mountain wave, rather than across a single model level.

The depth of the blocked flow layer is defined to be

$$z_b = \max(0, H(1 - F_{av}/F_c)). \tag{8}$$

Here, we introduce a depth average Froude number, $F_{av} = \bar{U}/(N_{av}H)$, where $\bar{U}$ is the speed of the horizontal wind resolved in the direction of the low-level flow averaged from $z = H/2$ to $H$, and $N_{av}$ is the buoyancy frequency averaged over a depth $z_{av}$. Following Vosper et al. (2009), who showed that the stability above a mountain can have a significant effect on the drag

exerted within the blocked flow layer below, $z_{av}$ is defined as

$$z_{av} = \max(H, z_n) + U_{av}/N_{av}, \tag{9}$$

where $z_n$ is the depth of a near-surface neutral layer (if present) and $U_{av}$ is the depth averaged wind speed from the surface to $z = z_{av}$. This empirical expression for $z_{av}$ was obtained from numerical simulations designed to examine the effect on the drag of neutral stability (as might typically be found in a well mixed boundary layer) below the mountain summit. The buoyancy

frequency $N_{av}$ is defined as a bulk average over the depth $z_{av}$ and thus depends on the difference in potential temperature, $\Delta\theta$, between $z = z_{av}$ and the surface i.e. $N_{av}^2 = (g/\theta_{av})\Delta\theta/z_{av}$, where $\theta_{av}$ is the mean potential temperature below $z_{av}$. Since the inputs required to solve Eq. (9) are themselves depth averages, the equation must be solved iteratively.

   In common with the Lott and Miller (1997) scheme a wave saturation approach is used to determine where gravity-wave drag is exerted. Wave breaking is assumed to take place when the local non-dimensional wave amplitude, $\eta N/U$ (where $\eta$ is

the vertical displacement associated with the gravity wave), exceeds a critical value, $\eta_{sat}$. When this occurs a proportion of the wave stress is exerted on the flow and the wave amplitude is reduced accordingly such that $\eta N/U = \eta_{sat}$. The wave drag is applied over a depth proportional to a hydrostatic vertical wavelength,

$$\lambda_z = 2\pi U(z)/N(z) \tag{10}$$

centred on the level of wave breaking, where $\lambda_z$ is constrained to lie within a range of values ($250\,\mathrm{m}$ and $3\,\mathrm{km}$). Applying

the wave drag in this way is consistent with the findings of Epifanio and Qian (2008) (see their Fig. 12) who showed that, in an ensemble of simulations of low-level wave breaking, stress deposition occurred over a range of depths between a half and full vertical wavelength. The numerical stability of the scheme is also improved by applying the stress over more than a single model level.

   The expression for the drag in the blocked flow layer is identical to that specified by Lott and Miller (1997). The size of

the drag is proportional to the drag coefficient, $C_d$, which along with the critical Froude number, $F_c$, is treated as a tuning parameter. The expression for the mountain-wave stress is also identical to Lott and Miller's, other than the modification





required to account for the reduced cut-off mountain height, in which $H$ is replaced by the height of the mountain which protrudes above the blocked layer, $H - z_b$. The mountain-wave stress is proportional to the tuning parameter $G_s$. The final tuning parameter is the threshold non-dimensional saturation wave amplitude, $\eta_{sat}$. The values of these parameters used in GA6.0 are $C_d = 4$, $G_s = 0.5$, $F_c = 4$ and $\eta_{sat} = 0.25$. These were identified in testing as giving improved performance in

terms of global model errors in mid-latitude winds, surface pressure and geopotential heights. As shown by Vosper (2015) who compared the drag due to explicitly resolved processes in high resolution simulations of flow over the steep mountainous island of South Georgia with the parametrised drag at coarse resolution, the 5A scheme can be tuned to give a very accurate representation of the true surface pressure drag and gravity-wave stress. However, the parameters required to achieve optimal results for an individual mountain range are in general not the same as those which optimise global performance.

### 3.6 Non-orographic gravity wave drag

#### 3.6.1 Tuning the launch amplitude of the non-orographic scheme (GA ticket #124)

As discussed in Sect. 2.7, the simulation of a realistic tropical quasi-biennial oscillation (QBO) in the UM relies on momentum supplied by the spectral sub-grid non-orographic gravity wave scheme. Although this is notionally a "sub-grid" scheme, for the period of the model's QBO to match that observed in reality it must model the breaking of both sub-grid waves and those

on larger scales that have been unrealistically damped by other processes such as the model's semi-implicit off-centring or semi-Lagrangian advection scheme.

One major impact of the ENDGame dynamical core's reduced off-centring is that its semi-implicit time stepping damps wave activity in the model far less than the dynamical core used in GA4 and before. For gravity waves, the most illustrative examples of this come from the improved simulation of orographically-forced lee-wave clouds in high resolution model simulations. A

similar increase is seen in non-zero phase speed gravity waves, however, which requires a retune of the non-orographic scheme. The simplest approach is to tune the amplitude of the launched waves by adjusting the "launched spectrum scale factor" ($C_{l0}$), which has been reduced from $\sim 5.13 \times 10^{-9}\,\mathrm{s}^{-2}$ in GA4.0 to $\sim 4.10 \times 10^{-9}\,\mathrm{s}^{-2}$ in GA6.0. In a pair of 25 year atmosphere/land-only climate simulations at N96 and N216 resolution, the period of the QBO measured at $30\,\mathrm{hPa}$ is $32.3 \pm 4.6\,\mathrm{months}$ and $28.8 \pm 2.9\,\mathrm{months}$ respectively, compared to a value of $27.0 \pm 3.5\,\mathrm{months}$ from ERA-Interim, which reflects the fact that the

value of $C_{l0}$ was chosen by tuning the QBO in an N216 resolution simulation. The longer period at N96 is consistent with fewer resolved waves to deposit momentum in the stratosphere at this lower resolution, which suggests that our current approach of using the simple scheme with a single global value of $C_{l0}$ may need revisiting in future configurations in the context of the new dynamical core.





## 3.7 Convection

**Increased entrainment rate for deep convection (GA ticket #74)**

In GA6.0 we alter the entrainment rate for deep convection to use a vertical profile similar to that used in GA3.0, but with its magnitude increased by 25%. The motivation for this change is to improve the model's representation of the Madden–
Julian Oscillation (MJO, Madden and Julian, 1971), which is the dominant mode of tropical intraseasonal variability, where large-scale organised convection propagates from the Indian Ocean to the Pacific with its convective and dynamical signatures affecting weather patterns globally (see for example the review in Lau and Waliser, 2005). Despite its importance in the global climate system, the MJO is still poorly represented in state of the art climate models (Hung et al., 2013). Studies show that model representations of the MJO can be improved by changing specific aspects of their convection parametrisa-
tion schemes. Most models lack intraseasonal intermittency in their precipitation (Jia-Lin et al., 2008; Xavier et al., 2010) and changes that inhibit deep convection appear to be particularly effective in improving the MJO (Wang and Schlesinger, 1999; Maloney and Hartmann, 2001; Jia-Lin et al., 2008; Zhang and Mu, 2005; Kim and Kang, 2012). However, there has been an apparent conflict between a model's fidelity for the MJO and its fidelity for the mean state (Kim et al., 2011). Microphysical processes such as the entrainment rate can have significant impact on the properties of simulated convection. This could also
be relevant for large-scale processes such as interactions between moisture and convection, between convection and dynamics and between clouds and radiation, all of which have been suggested as being important for the MJO. In this section we present a test of the impact of entrainment and detrainment changes on the MJO, which was used to motivate the change in GA6.0.

In the UM, the entrainment rate is a pressure dependant function represented as

$$\epsilon = \alpha(P/P_*)^r, \tag{11}$$

where $\epsilon$ is the entrainment rate, $p$ is the pressure at model levels, $P_*$ is the surface pressure and $\alpha$ and $r$ are user input parameters. The mixing detrainment rate is related to the entrainment by

$$\delta = \alpha_{det}\epsilon(1 - RH)^2, \tag{12}$$

where $\delta$ is the detrainment and $RH$ is the relative humidity with respect to water (at temperatures above 0°C) or ice (at temperatures below 0°C) and $\alpha_{det} = 3.0$ for both GA4.0 and GA6.0. During the development of GA4.0 it was found that a 50%
increase in the GA3.0 deep entrainment profile from [$\alpha = 0.9$, $r = 1$] to [$\alpha = 1.35$, $r = 1$] resulted in improved MJO characteristics and significant reductions in tropical errors (tropical cyclones, South Asian monsoon, African Easterly Waves etc., Klingaman and Woolnough (2014); Bush et al. (2015)), but this change also increased model biases in the upper troposphere. Motivated by this, GA4.0 used a similar deep entrainment profile, but with $\alpha = 1.35$ and $r = 2$ in Eq. (11) (shown in Fig. 3a). This profile gave higher entrainment rates at lower model levels (black curve in Fig. 3a) and hence more low and mid-level
clouds to help feed the convective moistening in the recharge phase of MJO convection. The profile has low entrainment rates at upper levels, which were chosen to reduce the upper tropospheric cold biases introduced by the [$\alpha = 1.35, r = 1$] profile.





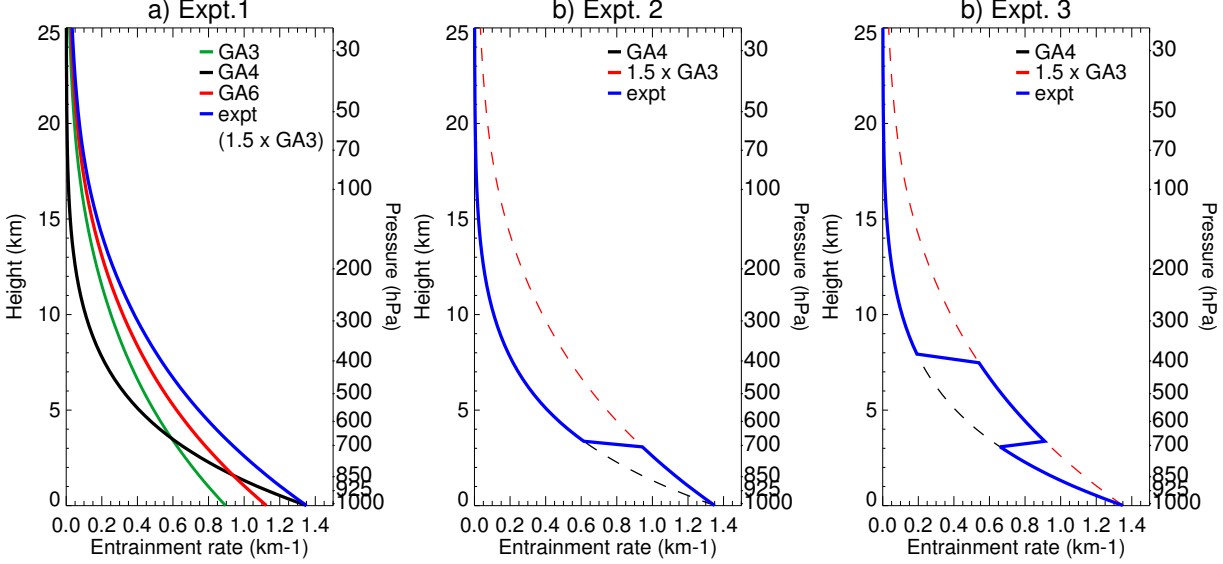

**Figure 3.** Examples of the idealised entrainment profiles for the experiments. a) shows the profiles of deep and mid-level entrainment in GA4 and the test profile (expt) with $[\alpha = 1.35, r = 1]$. b) shows the $[\alpha = 1.35, r = 1]$ profile for model levels under 700 hPa used in the second experiment. For third experiment $[\alpha = 1.35, r = 1]$ is maintained for model levels between 700 and 400 hPa (c), whilst the rest of the levels follow a $[\alpha = 1.35, r = 2]$ profile.

However this change did not lead to a significantly improved MJO simulation in GA4.0. This suggested the need to understand the role of low- and mid-level entrainment on the humidity profiles and the MJO.

We conduct a set of idealised experiments to understand the relative impact of higher entrainment in the lower and mid troposphere. Examples of the specified idealised entrainment profiles are shown in Fig. 3. In the first experiment (Fig. 3a), 5 $[\alpha = 1.35, r = 1]$ has been tested for the entire model column. For the second experiment (Fig. 3b), an $[\alpha = 1.35, r = 1]$ profile has been implemented for model levels under 700 hPa and for the third experiment $[\alpha = 1.35, r = 1]$ is maintained for model levels between 700 and 400 hPa (Fig. 3c), whilst the rest of the levels follow a $[\alpha = 1.35, r = 2]$ profile.

The process-based diagnostic we use to evaluate the convective moistening is the composite of RH profiles for different precipitation intensity bins (Fig. 4). This diagnostic has been shown to be useful to compare the moisture sensitivity of deep 10 convection in models with that in observations (Xavier, 2012). The average behaviour of the changes in RH transition from the low rainfall regime to the high rainfall regime is evident from Fig. 4a. The increase in RH in the mid-levels for moderate rainfall values is an indication of the convective moistening in the observations. The GA4.0 base line model has significant biases in representing this relationship (Fig. 4b) with the model producing much lower RH for low and high rainfall regimes.

The experiment 1 with increased deep entrainment (Fig. 3a, $[\alpha = 1.35, r = 1]$) introduces more moisture to the mid-levels 15 for moderate to intense precipitation intensities. A large part of the changes from this experiment is reproduced by experiment 3 (Fig. 3c) which has a higher entrainment rate between 700 and 400 hPa. These higher entrainment tests (1 and 3) produce an improved MJO amplitude of OLR compared to the GA4.0 control (not shown), which is confirmation that mid-level moisture



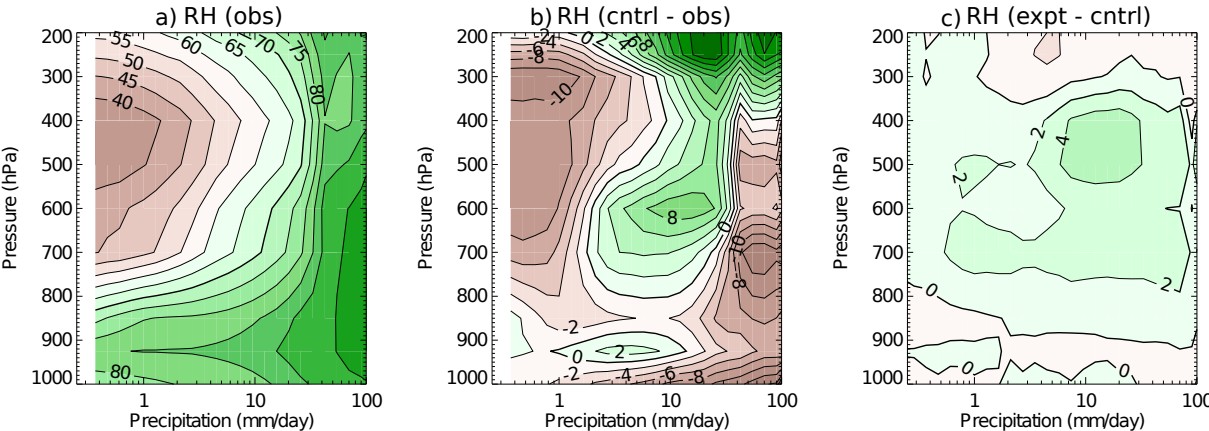

**Figure 4.** (a) Composite profiles of RH binned by daily average rain rate $(\mathrm{mm\,day^{-1}})$ over the Indian Ocean - west Pacific region (15°S-15°N, 50°-150°E) from ERA-Interim reanalysis (Berrisford et al., 2009, referred to as obs). The 70% RH countour is plotted with a thick line. (b) is the difference between the RH composite from obs and the GA4.0 control experiment with $[\alpha = 1.35, r = 2]$. (c) shows the difference in RH composites between the experiment with $[\alpha = 1.125, r = 1]$ (GA6.0 profile) and the control (GA4.0).

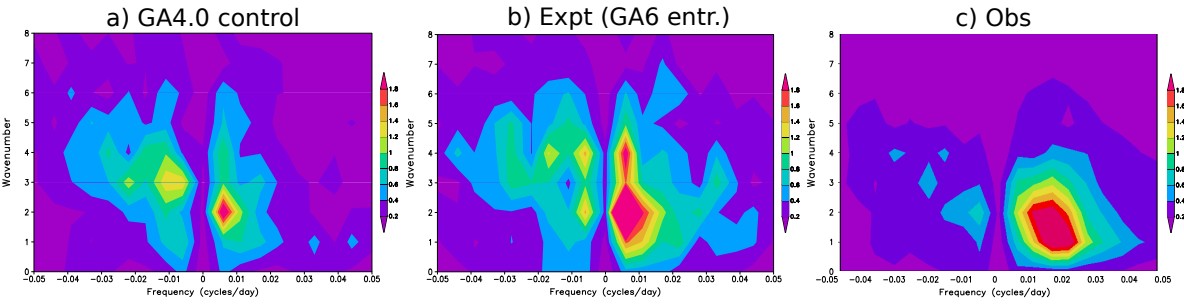

**Figure 5.** Wavenumber-frequency power spectra of boreal winter Outgoing Longwave Radiation from (a) GA4.0, (b) GA4.0 but with the GA6 entrainment profile and (c) from NOAA satellite estimates.

preconditioning is a critical element in improving the modelling of the MJO and explains why the approach attempted in GA4.0 was not successful in doing so.

As a result of the higher entrainment, however, the convective plumes have a tendency to terminate at a lower level, which can have a detrimental effect on the upper tropospheric temperature biases (not shown). Therefore in order to find a balance between including the MJO and other tropical phenomena discussed above and limiting any increase in upper tropospheric temperature biases, an intermediate entrainment profile with $[\alpha = 1.125, r = 1]$ has been chosen for GA6.0 (shown as the red line in Fig. 3a). Fig. 5 shows the wavenumber-frequency power spectra of Outgoing Longwave Radiation (OLR) from the GA4.0 baseline, the GA6.0 profile with $\alpha = 1.125, r = 1$ and from the NOAA satellite observations. The eastward MJO power has





been significantly improved in the 10-90 day band in the experiment compared to the GA4.0 baseline. There is no substantial reduction in the westward power for equatorial Rossby waves, however, unlike in the experiments with $\alpha = 1.35, r = 1$.

**Safety checks in the convection scheme (GA ticket #49)**

An investigation of some numerical model failures with GA3 and GA4 revealed a few areas of unsafe code in the convection

scheme. A series of changes known as the "convection safety checks" were introduced to prevent such problems. At present, the convection scheme works on profiles valid part way through a model time step, which can contain small, negative moisture values. There are already checks to stop the convection scheme seeing negative profiles of cloud condensate, so this change adds checks to prevent the convection scheme seeing negative water vapour.

In GA configurations, the convection scheme is sub-stepped, i.e. there are two calls to convection per model time step.

Sometimes, the shallow or deep scheme fails to convect, often on the second sub-step, but still produces an increment to the prognostic fields. There are also some cases where the deep or mid-level convection scheme fails to convect properly, producing an ascent with negative CAPE. Failed or unrealistic convective ascents are now prevented from incrementing the model prognostics.

### 3.8 Atmospheric aerosols and chemistry

**Improved treatment of the indirect aerosol effect when using aerosol climatologies (GA tickets #32 and #65)**

In GA3.0 and GA4.0 simulations that do not include the prognostic aerosol scheme, the direct aerosol effect (i.e. the reflection, absorption and scattering of radiation by the aerosol itself) is treated with the same method as in prognostic aerosol simulations, but uses three-dimensional speciated climatological aerosol masses rather than masses from the prognostic scheme. This gives a realistic spatial and temporal representation of the aerosol fields, but also ensures that the interaction of these climatologies with

the radiation scheme is identical to that in prognostic aerosol simulations, which ensures traceability between these different implementations of the GA configuration. For the indirect effects (i.e. the impact of the aerosol on the number and hence the radiative impact/properties of cloud droplets and the impact of the number of droplets on their size and hence the removal of moisture and clouds through precipitation), an extremely simple approach was adopted. This assumed a fixed potential droplet number concentration of $100\,\mathrm{cm}^{-3}$ over model sea points (representing relatively clean maritime air masses) and $300\,\mathrm{cm}^{-3}$

over land points (representing more polluted continental air). This was shown by Mulcahy et al. (2014) to lead to large cloud and radiation biases, particularly in clean air regions over land such as northern Canada, where the assumed aerosol loadings are considerably too high.

In GA6.0 we address this by extending the use of our speciated aerosol climatologies to the indirect aerosol effects. We do this by combining the climatologies already used for the direct effect with the parametrisation of Jones et al. (1994, 2001)

already used in prognostic aerosol simulations to provide a climatological potential cloud droplet number to be used by the radiation and the microphysics schemes.



**Reverted roughness lengths used for aerosol dry deposition (GA ticket #63)**

As discussed in Sect. 4.5.2 of Walters et al. (2014), another known problem in GA4.0/GL4.0 was that the changes to the ratio of surface roughness lengths for heat/moisture to those for momentum ($z_{0h}/z_{0m}$) listed in Table 3 of that publication had an unexpected impact on aerosol deposition. In particular, the increase in $z_{0h}$ for trees to be larger than $z_{0m}$ by-passed the resistance

5 to exchange from the laminar flow layer such that, over forested tiles, aerosols were deposited far too easily. This has been rectified in GA6.0/GL6.0 by removing the direct link between heat/moisture exchange and aerosol deposition and introducing an additional roughness length, $z_{0,\text{CLASSIC}}$, that is only used in deposition of prognostics in the "CLASSIC" aerosol scheme. The ratio $z_{0,\text{CLASSIC}}/z_{0m}$ for all surface types has then been reverted to the value of 0.1 that was used for heat and moisture prior to GL4.0. Figure 6 shows the impact of this change on the total aerosol optical depth (AOD) during September–November

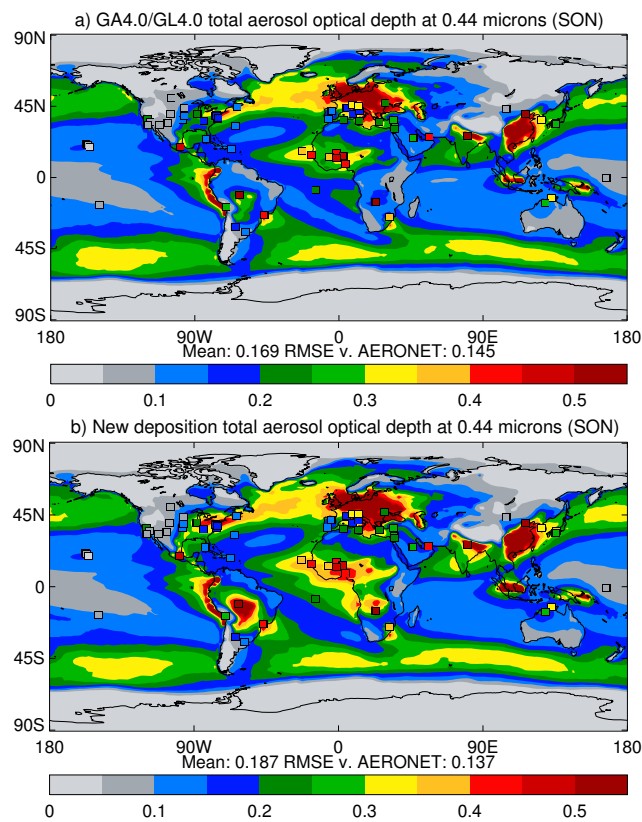

**Figure 6.** Mean total aerosol optical depth during SON from (a) a 10 year N96 atmosphere/land-only climate simulation using GA4.0/GL4.0 and (b) an equivalent simulation using $z_{0,\text{CLASSIC}}/z_{0m} = 0.1$ compared to climatological values from the AERONET sun photometer network (filled squares in both plots).

10 (SON) in a 10 year N96 atmosphere/land-only climate simulation, where the filled contours show the values from the model and the filled squares show the equivalent fields from the climatology of the AERONET sun photometer network (Holben et al.,





1998). As expected, the largest impact can be seen in the forested areas of central Africa and South America, where during this season the production of biomass burning aerosol reaches its peak. An investigation of the aerosol budget confirms that in the GA4.0/GL4.0 control, the majority of biomass burning aerosol is deposited back to the surface before it is transported away from its original source, whilst with the reduced $z_{0,\mathrm{CLASSIC}}$, the increased resistance to deposition allows more remote transport

and hence a larger average loading. The plot shows that locally this improves the agreement with the observations whilst away from these regions there is little impact on the aerosol loading. Globally, the root-mean-square (RMS) error in AOD is reduced by about 5%.

### 3.9 Land surface and hydrology: Global Land 6.0

**Improved treatment of the surface albedo (GA ticket #96 and GL ticket #8)**

JULES models the albedo of the land surface by specifying an individual albedo for each surface tile at each grid box. In GL4.0, the snow-free albedo of bare soil is spatially varying using the climatology of Houldcroft et al. (2008), whilst for each of the other surface types we use a single global value fitted to this dataset via the approach described in Brooks et al. (2011). For vegetated tiles, this is combined with the bare soil albedo and the leaf area index according to Monsi and Saeki (1953) to account for seasonally-varying vegetation; each tile's albedo is then updated further in the presence of snow.

The spatial variability of the albedo is well observed and whilst the approach used in GL4.0 can reproduce these observations reasonably well, there are still limitations to using a single value for the snow-free albedo for each surface type. In GL6.0, we improve on this approach by using a climatological snow-free albedo based on the GlobAlbedo dataset of Muller et al. (2012). In order to preserve contrast between the different surface types, we combine this climatology with the current approach by calculating a "first guess" albedo in each grid box using the same method as in GL4.0; the snow-free albedo of each tile is then

scaled (within limits to stop unrealistic values) until the grid box mean albedo best matches the value in the climatology. This maintains sensible differences between the tile albedos, but produces a final albedo which agrees well with observations.

Note that the approach of using an observed albedo is not suitable for climate change experiments that include a change in land usage. Such simulations should revert to the original approach of specifying an albedo for each surface type, but we recommend that the present day simulations are used as a benchmark with which to improve on the values used in Brooks et al.

25 (2011).

The impact of this change is to improve the surface energy budget of the model, which specifically improves near-surface temperature errors over continental land in the summer hemisphere. Figure 7 shows the impact of this change on the growth of temperature errors compared to screen observations over North American land in a set of 12 forecast case studies from the summers of 2011 and 2012 run at N320 resolution (approximately 40 km in the mid-latitudes) from independent (operational

ECMWF) analyses. This also shows the combined improvement from both the albedo climatology and the reduced (1 h) radiation time step discussed in Sect. 3.3.




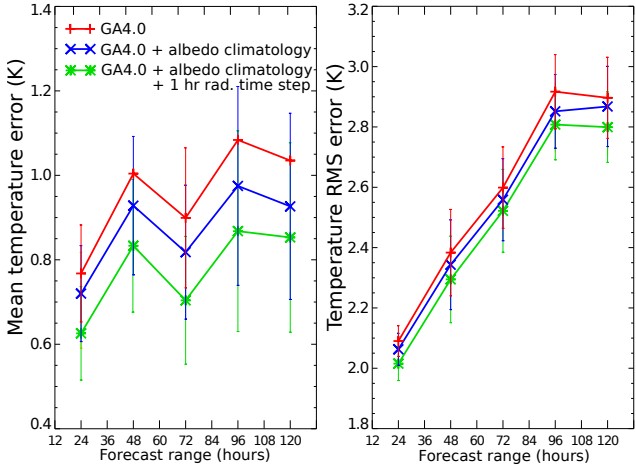

**Figure 7.** Mean bias (left) and RMS error (right) in screen-level temperature vs observations over North American land from a set of 12 N320 resolution forecast case studies from summer 2011 and 2012 run from operational ECMWF analyses.

### Changes to the roughness length of sea ice (GL ticket #32)

As documented in Walters et al. (2011), the GL3.0 "trunk" configuration on which both GL4.0 and GL6.0 are built and the GL3.1 "branch" configuration used for operational global NWP used very different values for the roughness length of sea ice. In GL3.1 we used a momentum roughness length of 3.2 mm for pack ice and 100 mm for marginal ice, whilst in GL3.0 we

assumed a roughness length of 0.5 mm for both. GL3.1 used the original default values in first the UM and then the JULES code base, that were never altered in operational NWP. The values used in GL3.0 had been previously tuned to improve the simulation of sea ice flow in a previous coupled climate configuration of the UM (McLaren et al., 2006).

Experimental determinations of the roughness length of sea ice have been performed at only a few locations and have yielded varying results. What evidence there is, however, suggests that drag coefficients were underestimated in GL3.0, but

overestimated in GL3.1. Weiss et al. (2011) report measurements over the Weddell Sea and suggest roughness lengths of 0.45 mm for young ice and 4.1 mm for pack ice. For marginal ice, Andreas (2011) compile the measured drag coefficients from various studies. Mostly, the drag coefficient lies in the range 0.001–0.0025 (corresponding to roughness lengths of ≈0.03– 3.35 mm), although a few observations of drag coefficients of 0.004 (roughness length ≈18 mm) have been reported.

Global NWP trials using data assimilation show improved verification of southern hemisphere winds and sea-level pressure

with the larger GL3.1 values whilst coupled climate simulations show only a small sensitivity of the climatological sea ice simulation, so pragmatically we have adopted the GL3.1 settings in GL6.0. We will further investigate these settings in the development of future configurations.



## 3.10 Ancillary files and forcing data

The only significant change to ancillary files in GA6.0 is the inclusion of the new snow-free land-surface albedo ancillary, derived from the GlobAlbedo dataset described in Muller et al. (2012), which is required for the improved treatment of the land surface albedo discussed in Sect. 3.9.

## 4 Differences between Global Atmosphere/Land 6.1 and Global Atmosphere/Land 6.0

As with previous GA configurations, the operational implementation of GA6/GL6 in the Met Office operational global NWP system includes a small number of scientific differences from the GA6.0/GL6.0 "trunk", although the number of these differences has significantly reduced since GA3.1/GL3.1 (Walters et al., 2011). For completeness, however, we recognise this by defining this as a "branch" configuration, which we label GA6.1/GL6.1. These differences are documented below.

## 4.1 Convection

### 4.1.1 CAPE closure timescale

The UM's timescale for CAPE closure is a parameter that has received a lot of attention in recent rounds of model development. Using a longer CAPE timescale has been shown to reduce the spatio-temporal intermittency of the UM's convection scheme by reducing its tendency to remove most of the convective instability in a single timestep, which in turn can improve the mean-state representation of regional phenomena such as the climatological south Asian monsoon. Using a shorter CAPE timescale, however, improves the modelling of more intense tropical systems such as tropical cyclones and improves the short-range extra-tropical prediction skill of the model. This latter point is illustrated in Table 3, which shows the reduction in forecast errors from reducing the CAPE timescale from 1 h to 30 min in a set of 24 forecast cases. The reduction in RMS errors is small, but almost always beneficial and is achieved without affecting the variability of the forecast as measured by the standard deviations (not shown). Similar results have been found in full data assimilation trials run over multiple periods and with multiple baseline configurations.

The CAPE timescale of 1 h used in GA6.0 was chosen as a compromise between two extremes. Operationally, however, it has been hard to justify the small but consistent reduction in predictability associated with increasing the CAPE timescale from the previously operational value of 30 min used in GA3.1. For this reason, the GA6.1 configuration used for operational global NWP continues to use this shorter CAPE timescale. Our belief is that the lack of a single parameter value suitable for all purposes exposes a weakness in the current parametrisation and suggests that an alternative approach is required, such as a dynamically diagnosed CAPE timescale or an alternative convective closure.



| Parameter | RMS error (GA6.0) | RMS error (GA6.0 + 30 minute CAPE ts) | % diff |
|---|---|---|---|
| T+24 NH PMSL (hPa) | 1.408 | 1.405 | -0.2 |
| T+48 NH PMSL (hPa) | 1.813 | 1.800 | -0.7 |
| T+72 NH PMSL (hPa) | 2.405 | 2.380 | -1.1 |
| T+96 NH PMSL (hPa) | 3.287 | 3.261 | -0.8 |
| T+120 NH PMSL (hPa) | 4.123 | 4.100 | -0.6 |
| T+24 NH $\Phi_{500\,hPa}$ (dm) | 1.359 | 1.350 | -0.7 |
| T+48 NH $\Phi_{500\,hPa}$ (dm) | 1.751 | 1.738 | -0.7 |
| T+72 NH $\Phi_{500\,hPa}$ (dm) | 2.353 | 2.330 | -1.0 |
| T+24 NH $x\boldsymbol{v}_{250\,hPa}$ ($\mathrm{m\,s^{-1}}$) | 5.117 | 5.069 | -0.9 |
| T+24 SH PMSL (hPa) | 1.238 | 1.235 | -0.2 |
| T+48 SH PMSL (hPa) | 1.601 | 1.605 | +0.2 |
| T+72 SH PMSL (hPa) | 2.193 | 2.182 | -0.5 |
| T+96 SH PMSL (hPa) | 2.810 | 2.765 | -1.6 |
| T+120 SH PMSL (hPa) | 3.689 | 3.602 | -2.4 |
| T+24 SH $\Phi_{500\,hPa}$ (dm) | 1.387 | 1.363 | -1.8 |
| T+48 SH $\Phi_{500\,hPa}$ (dm) | 1.725 | 1.699 | -1.5 |
| T+72 SH $\Phi_{500\,hPa}$ (dm) | 2.160 | 2.120 | -1.9 |
| T+24 SH $\boldsymbol{v}_{250\,hPa}$ ($\mathrm{m\,s^{-1}}$) | 5.606 | 5.538 | -1.2 |

**Table 3.** The difference in root mean square error vs observations in a number of extra-tropical performance measures due to reducing the CAPE timescale from 1 h to 30 min in a set of 24 N320 resolution forecast case studies run from operational ECMWF analyses. The parameters are pressure at mean sea level vs synoptic observations (PMSL) and geopotential heights and vector wind errors vs radiosondes at 500 hPa and 250 hPa respectively ($\Phi_{500\,hPa}$, $\boldsymbol{v}_{250\,hPa}$).

## 4.2 Land surface and hydrology: Global Land 6.1

### 4.2.1 Aggregated surface tile

In addition to the CAPE timescale, another long-standing difference between operational global NWP and other operational configurations of the UM is that the former (including GL3.1) has always performed its land surface calculations over a single
5 land surface tile with the aggregated properties of the 9 individual surface types rather than performing these in parallel and aggregating the resulting fluxes. Initial investigations have shown that this is due to the Bowen ratio (i.e. the ratio of sensible to latent heating at the land surface) being higher in the 9 tile model, leading to large near-surface warm biases and near-surface low pressure biases in some regions during local summer.

It is not yet clear whether the "improved" performance of the aggregated tile is due to a deficiency in the 9 tile approach
10 (possibly due to errors in the specification of surface parameters) or due to some aspect of the global NWP system having

been developed to perform well with a 1 tile model (e.g. the details of the land surface data assimilation). In the absence of having made progress in understanding this issue, therefore, GL6.1 continues to use the aggregated tile approach that was used operationally with GL3.1. Because the aggregated tile approach is incompatible with holding snow on the vegetation canopy and with the use of the "inland water canopy" for modelling lakes, these schemes are also dropped from GL6.1. Finally, it

is impossible to sensibly aggregate the thermal and momentum roughness lengths (respectively labelled $z_{0h}$ and $z_{0m}$) using the range of values of $z_{0h}/z_{0m}$ from Table 3 of Walters et al. (2014), so in GL6.1 the value of $z_{0h}/z_{0m}$ for broadleaf and needle-leaved trees is reduced from the GL6.0 value of 1.65 to the GL3.0 value of 0.1.

### 4.2.2 Thermal conductivity of sea ice

Rae et al. (2015) describes the development of the Global Sea Ice 6.0 (GSI6) configuration of the Los Alamos CICE sea ice
model (Hunke and Lipscombe, 2010), which was developed in parallel to GA6.0/GL6.0 for use in coupled simulations as part of the Global Coupled model 2.0 (GC2) configuration (Williams et al., 2015). For consistency between the Global Land configuration in coupled and uncoupled simulations, where changes to GSI6.0 included changes to the JULES land surface model, we have included these same changes in our GA/GL simulations.

For one set of parameters, namely the thermal conductivity of sea ice and snow on top of sea ice (labelled $\kappa_{ice}$ and $\kappa_{snow}$
respectively) we omitted to make these changes in pre-operational NWP tests of GA6.1/GL6.1. Rather than fixing this issue, which would have required an additional round of trialling and a delay to operational implementation, we decided to include this change in the definition of GL6.1. The values of these parameters are shown in Table 4. As the presence of this difference

| Parameter | GL6.0 (& GSI6.0) | GL6.1 |
|---|---|---|
| $\kappa_{ice}$ | $2.63\,\mathrm{W\,m^{-1}\,K^{-1}}$ | $2.09\,\mathrm{W\,m^{-1}\,K^{-1}}$ |
| $\kappa_{snow}$ | $0.50\,\mathrm{W\,m^{-1}\,K^{-1}}$ | $0.31\,\mathrm{W\,m^{-1}\,K^{-1}}$ |

**Table 4.** Thermal conductivity of sea ice in GL6.0 and GL6.1.

was accidental, this will be removed in the next Global Land release. With prescribed sea ice fractions and thicknesses, the impact of these differences on an uncoupled GA/GL simulation are small, but non-zero. This is because the sea ice in these
simulations is specified with a fixed temperature at ice base, such that the sea ice surface temperature is dependent on its thermal conductivity. As shown in Fig. 8, in the winter hemisphere, where the near-surface air temperature is much colder than the freezing point of sea water, the reduced thermal conductivity in GL6.1 leads to a warmer surface temperature over sea ice. In the summer hemisphere (not shown), where the thermal gradient through the sea ice is much smaller, there is very little difference in the ice surface temperatures between the two configurations.





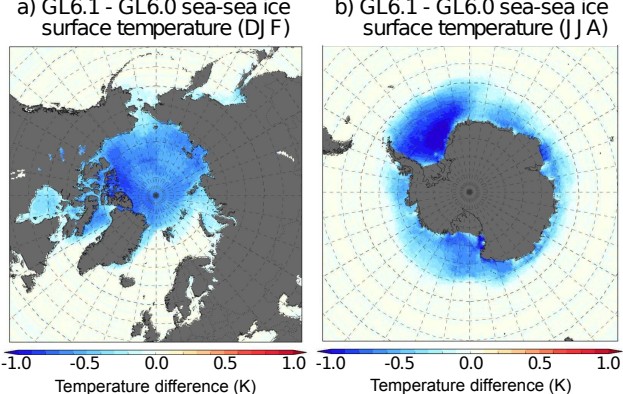

**Figure 8.** The difference in mean day 1 surface temperature over sea ice between GL6.1 and GL6.0 in sets of 12 N320 resolution forecast case studies run from operational ECMWF analyses in (a) December–February 2010/11 and 2011/12 (DJF) and (b) June–August 2011 and 2012 (JJA).

## 5   Model evaluation

In this section we illustrate the combined impact of the GA5 and GA6 changes on model performance. For most systems, the baseline used is the last documented configuration of GA4, but for NWP forecasts, we compare the GA6.1 configuration with the previous operational configuration of GA3.1. The difference between these operational systems includes the impact of changes in GA4, but we will not focus on these here. On implementing the NWP upgrade, we also upgraded the resolution of the deterministic NWP forecasts from N512 (approximately 25 km in the mid-latitudes) to N768 (approximately 17 km) and we include this impact in some figures where relevant.

### 5.1   Extra-tropical and tropical variability

The largest impact of the ENDGame dynamical core is the reduced implicit damping that comes from the reduced off-centring in its semi-implicit time stepping. As discussed in Sect. 3.1, for "New Dynamics" to remain numerically stable, its time stepping was set to be more implicit, which had the impact that previous GA configurations could not maintain sufficient mid-latitude variability. Figure 9 shows the global mean eddy kinetic energy (EKE) from a set of three-day forecasts as a function of horizontal resolution in GA configurations before and after the inclusion of ENDGame. With ENDGame (GA5), the EKE is increased in all resolutions and the ENDGame simulation at N216 (approximately 60 km in the mid-latitudes) displays higher EKE than N768 New Dynamics (GA4). The difference in EKE between different resolutions between N216 and N768 in GA5 is much smaller than in GA4 and the value is very close to the regridded verifying ECMWF analyses by N512. At N96 resolution (approximately 135 km in the mid-latitudes), earlier configurations used ECMWF "quasi-cubic" rather than cubic horizontal interpolation for the departure point in the semi-Lagrangian advection scheme (illustrated in Fig. 2 of Ritchie et al., 1995). This was originally introduced for computational efficiency and numerical stability, but also had the effect of increasing



the EKE. Moving to ENDGame has permitted the use of the more accurate cubic interpolation at this resolution, bringing it into line with higher resolution simulations, whilst maintaining the EKE so as to be comparable to its previous level. However, this does mean that N96 climate simulations do not exhibit the same increase in EKE with the upgrade to GA6 that is seen at other resolutions. Also, this increases the difference in variability between this resolution and the resolutions above.

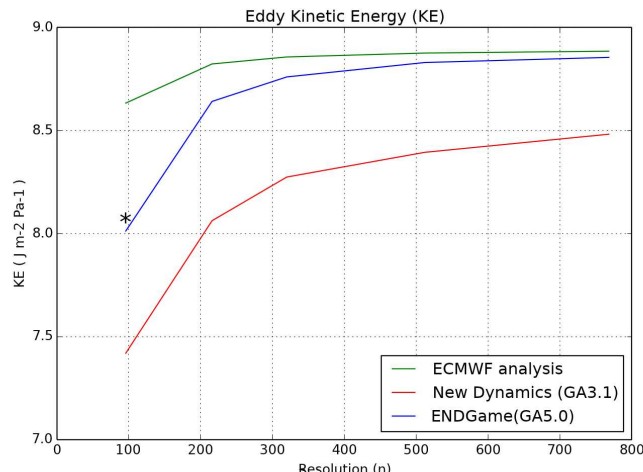

**Figure 9.** Eddy kinetic energy from sets of 12 three-day forecasts run from ECMWF analyses as a function of resolution compared to the verifying analyses reconfigured to those resolutions. Cubic interpolation of the departure point is used in the simulations shown in this plot, however the black asterisk marks GA4 at N96 with quasi-cubic interpolation.

Consistent with the loss of EKE in earlier configurations, Froude (2010) showed a drop in the intensity of extra-tropical cyclones through the forecast in GA3.1, which we demonstrate here in Figure 10. At N512, this is largely addressed by the inclusion of ENDGame in GA6.1 with a subsequent horizontal resolution increase to N768 having little additional impact. This shows some sign that cyclones in GA6.1 may be overly intense relative to analyses. This was also suggested by Mittermaier et al. (2015) who performed a different type of feature tracking, but came to similar conclusions: that cyclones

and jets had both become stronger in GA6 and are now occasionally too intense. Subsequent analysis has suggested that this over-intensification may be due to issues with the bias-correction of satellite data in the analysis, which are being addressed.

    In the tropics the most significant impact of GA6 is an improvement in the representation of tropical cyclones, which comes from a combination of ENDGame and the increased deep entrainment rate in the convection scheme. The benefits of this for short-range tropical cyclone forecasts (which include a 7% reduction in forecast track error for a given resolution) is

15 discussed fully by Heming (2016). Here, Figure 11 (reproduced from Heming (2016)) illustrates the improvement in tropical cyclone intensity. A marked weak bias in GA3.1 is considerably reduced in GA6.1, most notably at longer lead times (as shown by the reduced central pressure bias). A further improvement is gained from the increase in horizontal resolution. A good example of the changes in forecast intensity throughout the lifetime of a tropical cyclone is provided by Figure 12, which shows successive forecasts for the central pressure of Typhoon Bolaven — which made landfall over North Korea on





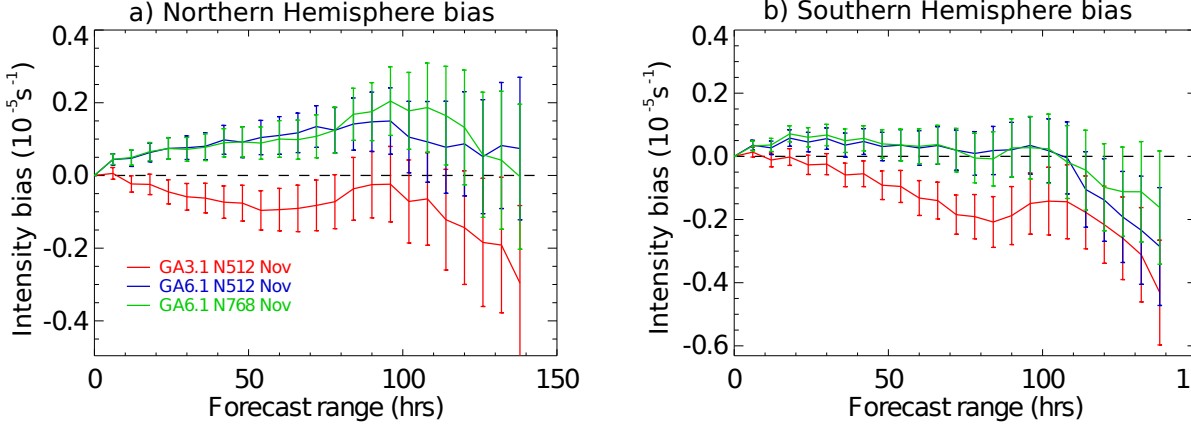

**Figure 10.** Bias in extra-tropical cyclone intensity (measured by $850\,\mathrm{hPa}$ relative vorticity) as a function of forecast lead time from data assimilation trials run through November–December 2012. This is obtained from cyclone tracking using Reading University's TRACK algorithm (Hodges, 1995). Red is the previously operational GA3.1. Blue and green show GA6.1 trials at N512 and N768 respectively.

$28^{\text{th}}$ August 2012 — compared to the official estimates of its "observed" pressure. GA6.1 has much deeper central pressures, and generally deepen at a comparable rate to what is observed. However, the pressures at the beginning of each subsequent forecast are not much deeper than in the control. This is consistent with the general weak bias at analysis time in Figure 11 and illustrates that the analysis cannot capture the intensities sustainable by the model and observed in the true system. Heming (2016) discuss subsequent changes to assimilate central pressures which have a positive impact on this analysis error. Another feature illustrated by this example, which is present primarily in tropical cyclones which move polewards into the subtropics, is the over-intensification towards the end of the forecast. This tends to occur in situations where the real cyclone loses intensity before landfall, which the model is usually unable to capture. One hypothesis for this is that the reduction in intensity in the real system is due to the cyclone removing heat energy from the upper levels of the ocean, and hence reducing a source of energy for further intensification. This process is not represented in the current NWP system, which uses a fixed sea surface temperature and hence a limitless source of heat energy.

Elsewhere, the spectrum of tropical variability has become richer, with the introduction of ENDGame particularly increasing eastward propagating Kelvin wave activity (Fig. 13). As noted above, the increased entrainment rate has also improved the MJO signal, although westward propagating Rossby waves with wavenumbers greater than 2 remain weak.

## 5.2 Surface weather

GA6 brings significant changes to the geographical distribution of climatological rainfall (Figure 14). Overall the spatial RMS error of this climatology is slightly reduced, however the existing dry bias over central and west Africa in June–August is exacerbated, as is the dry bias over the Maritime Continent in December–February (not shown).





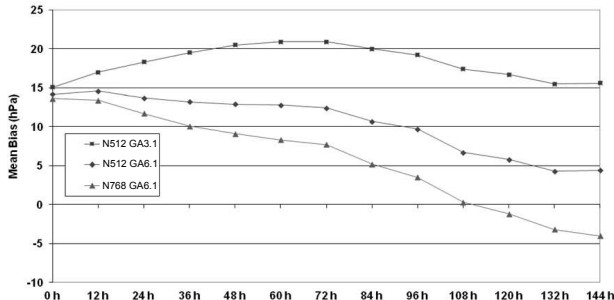

**Figure 11.** Mean tropical cyclone central pressure bias during data assimilation trials run from June–September 2012. Reproduced from (Heming, 2016), but with altered labels in the legend.

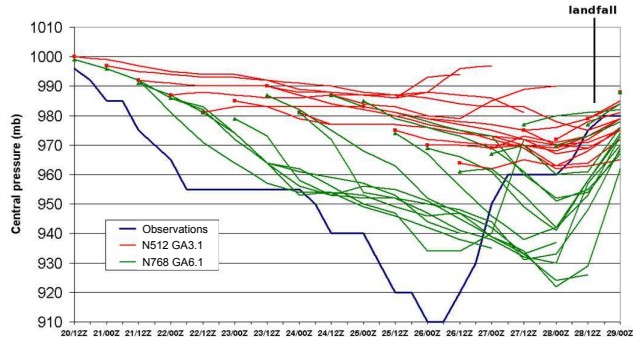

**Figure 12.** Central pressure evolution from successive forecasts for Typhoon Bolaven from data assimilation trials. The red lines are GA3.1 at N512 and green are GA6 at N768.

On climate timescales, the large dry bias over India persists, however on shorter timescales the distribution and variability of rainfall over India, such as precipitation associated with monsoon depressions, is improved (as illustrated in Fig. 15), increasing the utility of the model for NWP over the region.

In the mid-latitudes, precipitation associated with frontal features has become sharper and precipitation generally appears more organised with less spurious light rain. This is due to a combination of the increased intensity of fronts (i.e. another example of reduced damping with ENDGame) and physics improvements from GA4 (e.g. the improved representation of the drizzle size distribution). Subjective feedback from forecasters suggests that this is an improvement. Figure 16 illustrates these points and shows a case which resulted in disruptive heavy rain across southwest UK. In this case GA6.1 gave a signal for this event more than four days in advance, which compares with a signal given just over two days in advance from the control. Objectively, the reduction in spurious light rain is reflected in the SEEPS (Stable Equitable Error in Probability Space, Rodwell et al., 2010) score which is improved by 2% globally, mostly from situations which are forecast to have relatively light precipitation (compared with climatology) but are actually dry, particularly in the tropics.





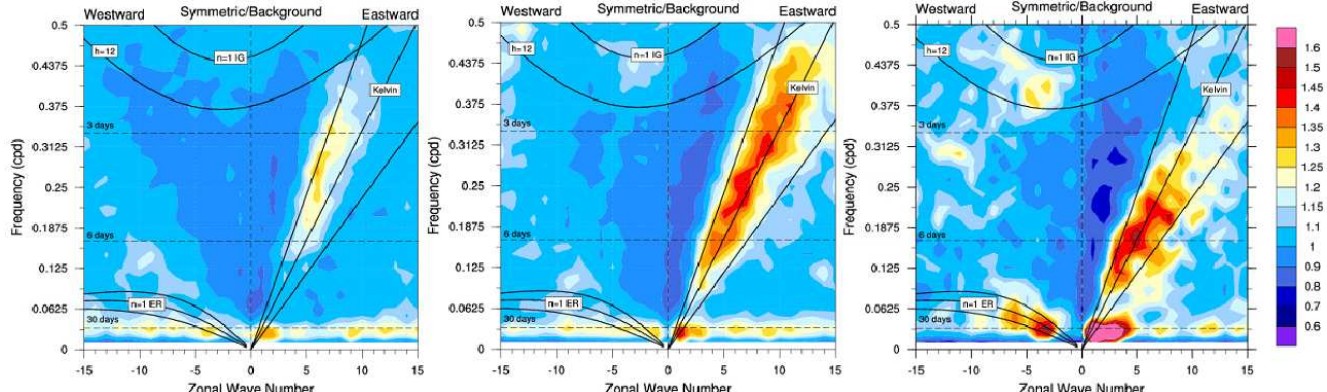

**Figure 13.** Tropical power spectra derived from surface precipitation fields between 15° S and 15° N (following Wheeler and Kiladis, 1999) for GA4.0 (left), GA6.0 (centre) and as observed from TRMM data (right, Huffman et al., 2007). Model data are from 20 year N96 atmosphere/land-only climate simulations.

Williams and Bodas-Salcedo (submitted) conduct a detailed evaluation of cloud in GA6 against a range of observational data and conclude generally good performance, although there is excess optically thin cirrus and boundary layer cloud is too optically thick. This generally good performance is reflected in the top-of-atmosphere radiation errors which are reduced in GA6 compared with GA4 (illustrated for the reflected shortwave radiation in Fig. 17). Most notably the overly reflective sub-tropical boundary layer cloud on the eastern side of ocean basins is reduced.

Additionally, GA6.1 delivers a global improvement in near-surface temperature errors due to the radiative improvements shown in Fig. 7 and the use of aerosol climatologies for the indirect aerosol effect and an improvement in near-surface wind errors (not shown) due to ENDGame's improved representation of fronts and cyclones and improvements from the 5A gravity wave drag scheme. These improvements are important as the increase in the resolution of global NWP models means they are increasingly used for surface weather prediction in addition to modelling the large-scale flow. In particular, the upgrade from N512 GA3.1 to N768 GA6.1 led to a 4-5% global increase in the Met Office near surface weather index, which includes the verification of screen-level temperature, near surface wind, precipitation, cloud amount, cloud base height and visibility. Over Europe, this means that the ≈ 17 km global NWP model now outperforms the previously operational 12 km limited area model, which is a justification for its retirement in 2014.

## 5.3 Mean error structure and large scale flow

Despite the large number of differences between GA6 and GA4, the mean tropospheric temperature structures of their model climatologies are broadly similar. In the stratosphere, however, GA6 is cooler away from the tropical tropopause region, as illustrated in Fig. 18. In contrast, GA5 is notably different with a very large warm bias of more than 6 K in the climatological mean at the tropical tropopause. This region includes the coldest temperatures that air parcels encounter during their ascent



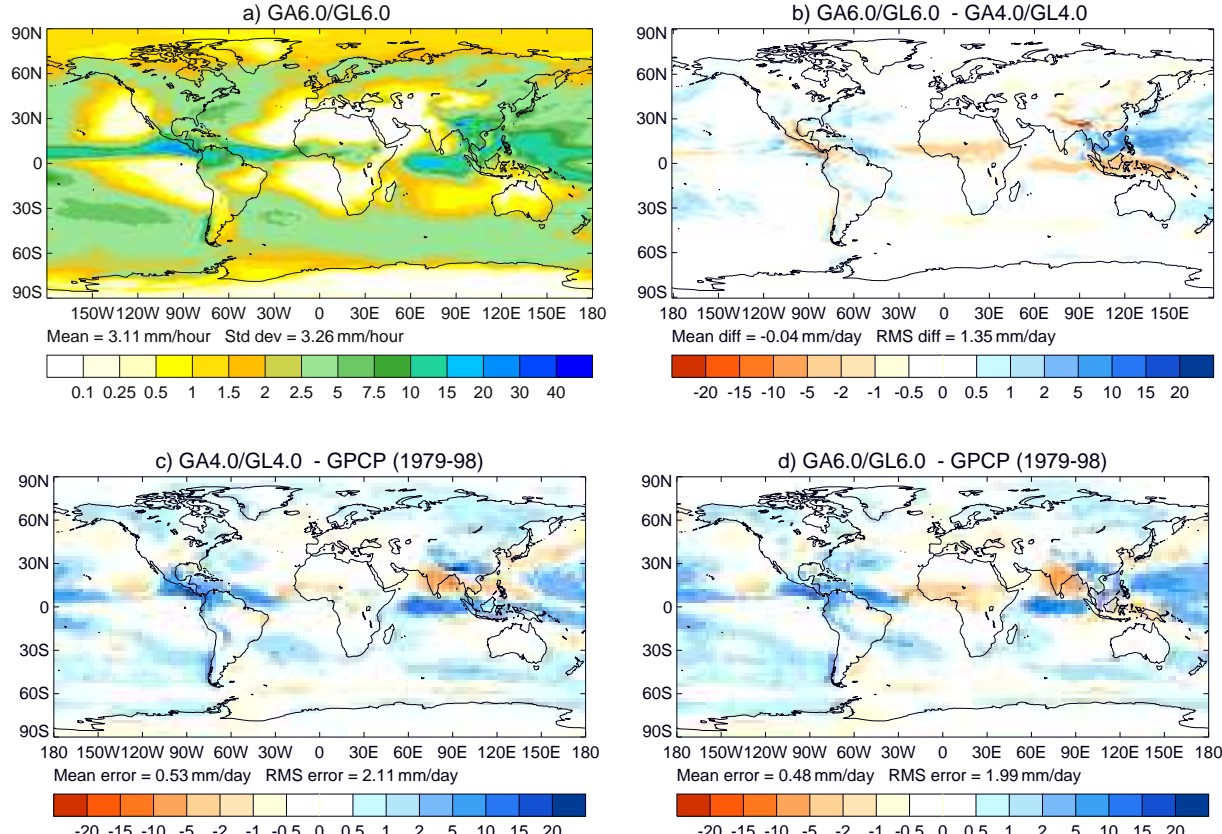

**Figure 14.** JJA precipitation rate (mm/day) in an N96 atmosphere/land-only climate simulation using GA6.0 (top left), the difference from GA4.0 (top right) and the bias against GPCP (Global Precipitation Climatology Project; Adler et al., 2003) for GA4.0 (bottom left) and GA6.0 (bottom right)

from the troposphere into the stratosphere, which itself limits the transport of moisture into the warmer regions of the strato-sphere (Fueglistaler et al., 2013; Zahn et al., 2014). This means that temperature biases in this region can lead to moisture biases throughout the stratosphere, which in turn will affect chemical processes simulated within Earth system models. The warm bias in GA5 was introduced by ENDGame's replacement of the "New Dynamics" non-interpolating in the vertical ad-vection of potential temperature with a fully three-dimensional semi-Lagrangian scheme, which in turn was alleviated in GA6 by using cubic Hermite rather than cubic Lagrange vertical interpolation for this variable. Although this interpolation change makes the advection scheme lower order and hence slightly less accurate in general, it is more accurate in regions of strong gradients, such at the tropopause (Hardiman et al., 2015).

For NWP runs, the upgrade from GA3.1 to GA6.1 has only a small impact on a basket of skill scores based on those exchanged between centres under the WMO's Commission for Basic Systems (CBS) to measure the accuracy of the large-scale flow (not shown). Since ENDGame increases the intensity of cyclones, fronts and jets, etc., in isolation this would tend

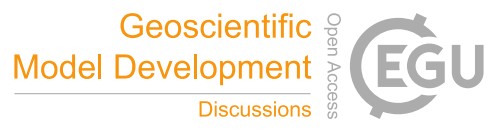

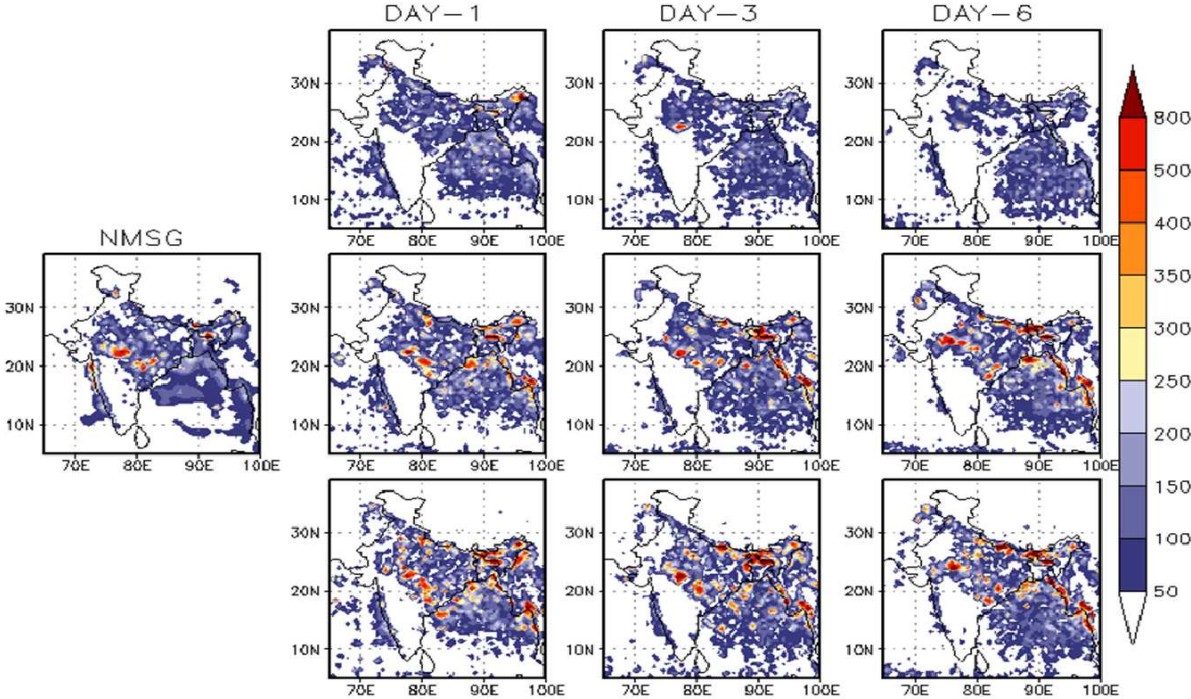

**Figure 15.** Variance of daily rainfall $(\mathrm{mm}^2\mathrm{day}^{-2})$ for a 72-day period during data assimilation trials in July–September 2012. The observations (on the left) are from the Indian National Centre for Medium Range Weather Forecasting (NCMRWF) Merged Satellite-Gauge (NMSG) product (Mitra et al., 2009). The model data to the right of this are from N512 GA3.1 forecasts (top), N512 GA6.1 (centre) and N768 GA6.1 (bottom).

to reduce scores due to a double penalty in calculating the RMS error in situations where the position of a feature is in error. Improvements in the accuracy of the forecasts from improvements in resolution, dynamics and physics changes alleviate this problem by offsetting the reduction from the more active dynamical core. One area in which NWP forecasts have deteriorated in the final package is in upper level tropical wind speeds (Figure 19). The wind speeds are increased, which reduces a negative

5 bias against observations, but results in an increased RMS error. The wind speed increase is a result of the combination of ENDGame, removing vertical diffusion in this region and increasing the convective entrainment rate, which are changes critical to other model improvements documented here.





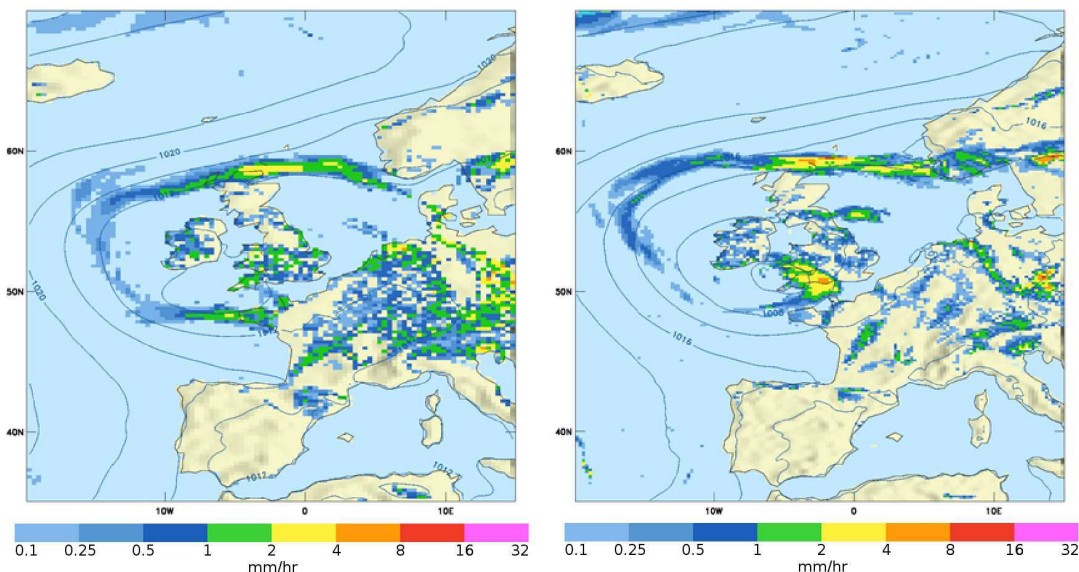

**Figure 16.** 108 hour forecast rainfall rates from the N512 GA3.1 (left) and N768 GA6.1 trial (right), valid 12 UTC 7th July 2012.

## 5.4    Problems identified with GA6.0/GL6.0

### 5.4.1    Problems with GA6 orography files

The Central Ancillary Programme code used to generate UM ancillary files for GA6 originally contained an error when rewritten for the ENDGame grid, which led to an $O(100\,\mathrm{m})$ "step" in the mean orography fields across the Greenwich meridian near the south pole and a localised flattening in the rows closest to the pole. This code has since been fixed and these errors removed; the resulting ancillaries will be officially part of GA7, but were also applied operationally in the Met Office global NWP suite on top of GA6.1 in August 2015.

### 5.4.2    Noise in the upper-level wind fields near the poles

High resolution global simulations using GA6.0 (i.e. simulations at a horizontal resolution of N512 and above) have exhibited problems with numerical noise in the meridional wind near the poles in the topmost few levels (i.e. at altitudes of $65\,\mathrm{km}$ and above). Usually, these are limited to the few rows closest to the pole, but during periods of strong upper-level cross-polar flow, this noise can be advected away from the pole and cause problems with model stability.

It is unclear whether the source of this noise is a feature of ENDGame, or whether it was also present in New Dynamics but removed by its aggressive polar filter. We have shown, however, that this noise can be significantly reduced by increasing



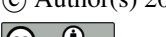

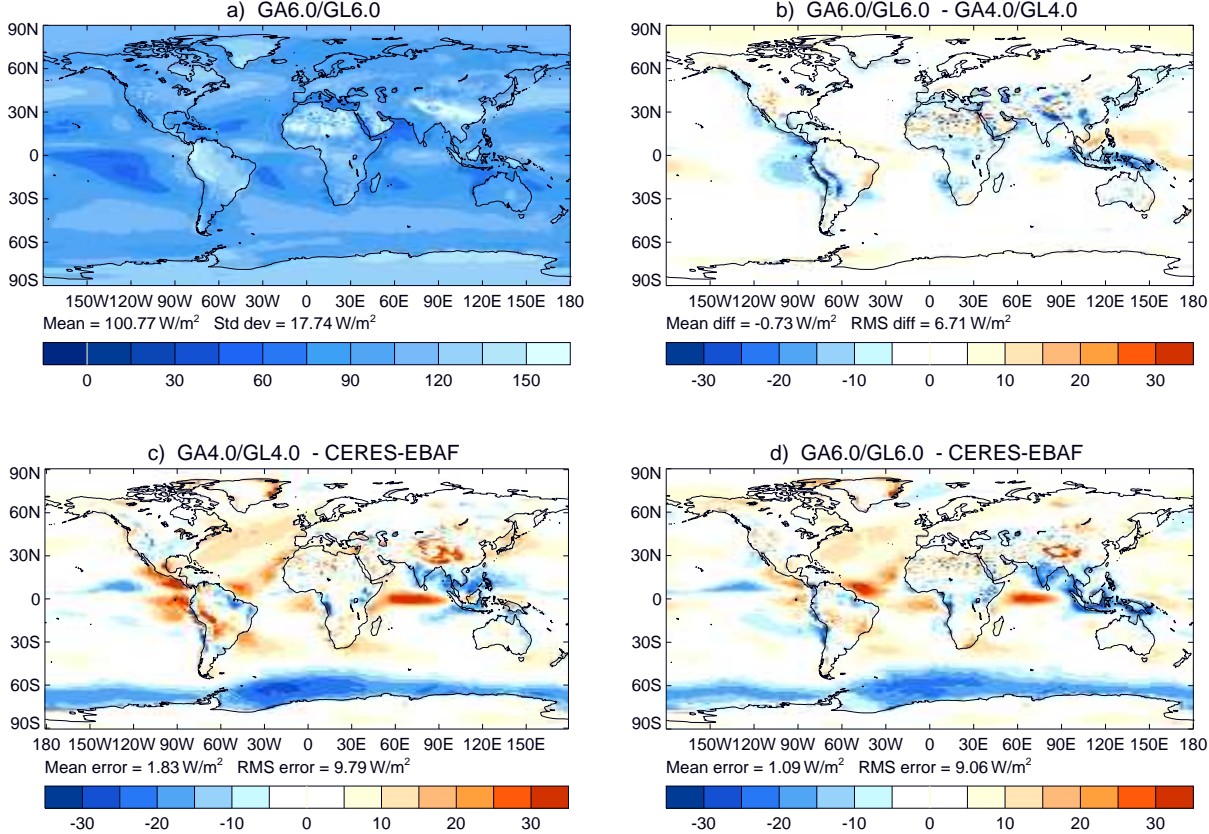

**Figure 17.** Top of atmosphere reflected shortwave radiation ($\mathrm{W\,m^{-2}}$) in N96 atmosphere/land-only climate simulations of GA6.0 and GA4.0 compared with CERES EBAF (Clouds and the Earth's Radiant Energy System–Energy Balanced and Filled dataset, Loeb et al., 2009). The layout is the same as in Fig. 14.

the accuracy to which the linear Helmholtz equation is solved. So in GA7.0 we will be reducing the "tolerance" used in the iterative Helmholtz solver by an order of magnitude. This change was also applied operationally in the Met Office global NWP suite on top of GA6.1 in August 2015.

### 5.4.3   Non-conservation of potential temperature

5   As discussed in Sect. 3.1, ENDGame requires a mass fixer run every time step to conserve the dry mass of the atmosphere. For some time, climate configurations of the UM have also used conservative advection algorithms such as those described in Priestley (1993) in the advection of moist prognostics to conserve total atmospheric moisture from one time step to another. Since the freeze of the GA6 configuration, we have found that ENDGame also requires a similar conservation algorithm applied to mass-weighted dry virtual potential temperature, which is otherwise not conserved. Climate models using GA6 will




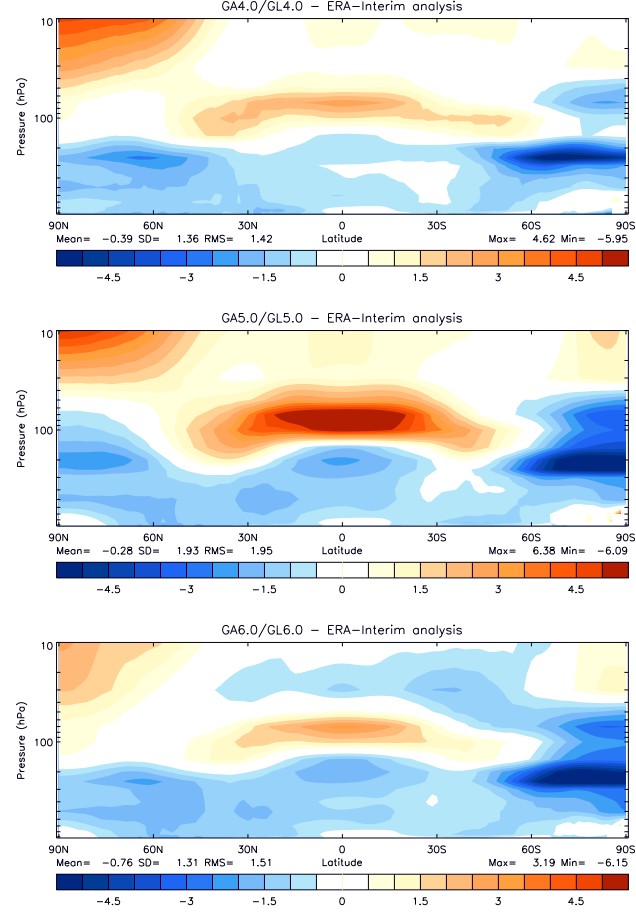

**Figure 18.** DJF zonal mean temperature bias in N96 atmosphere/land-only climate simulations compared to ERA-Interim. The panels from top to bottom are GA4.0, GA5.0 and GA6.0 respectively.

still enforce the conservation of energy by application of a daily global energy correction step, but this error can still lead to localised heating errors such as those observed at and around the tropical tropopause (Hardiman et al., 2015). For this reason, this error will be addressed in GA7.

## 6 Summary and conclusions

5   The inclusion of the ENDGame dynamical core is an important upgrade to the Global Atmosphere configuration of the UM. ENDGame maintains the benefits of "New Dynamics", whilst improving on its accuracy, stability and scalability. The improved accuracy significantly reduces the model's implicit damping, leading to a beneficial improvement to various modes of variability, such as the depth of extra-tropical cyclones and the definition of frontal systems. The improved stability now allows



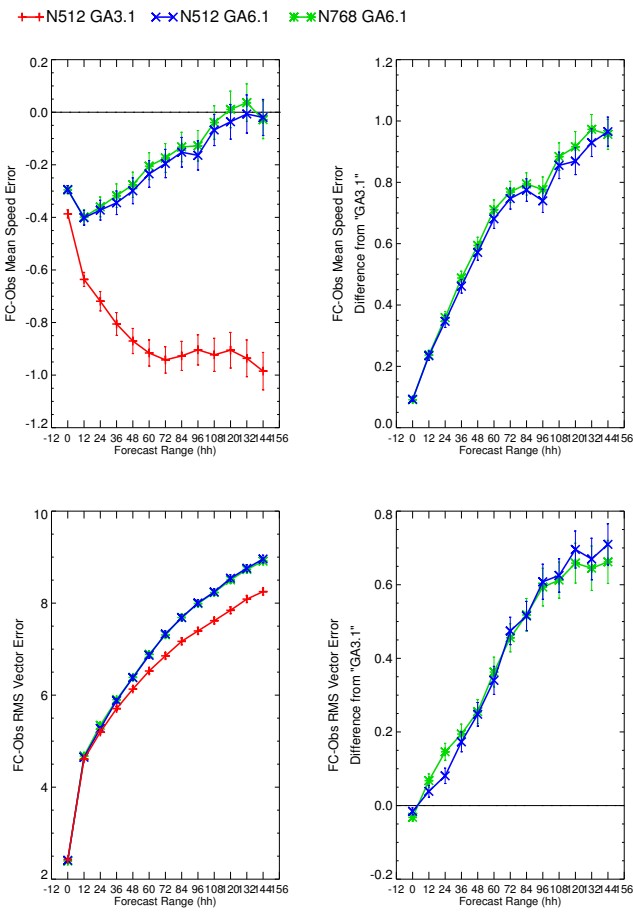

**Figure 19.** 250 hPa tropical winds from the GA3.1 (red) and GA6.1 at N512 (blue) and N768 (green) compared with radiosondes. Mean wind speed bias (top left); difference from GA3.1 (top right); RMS error (bottom left); difference in RMS error (bottom right).

us to perform high resolution climate simulations (at resolutions of N512 and above) for hundreds of years without experiencing model failures and the improved scalability means that we can continue to upgrade the resolution of the deterministic global NWP model over the next few years by taking advantage of the increasing number of processing cores in modern supercomputers[4]. The physics upgrades developed and implemented alongside ENDGame have further improved modes of tropical variability such as tropical cyclones and the MJO and led to improvements in the model's representation of surface weather.

---

[4]Despite the advances from ENDGame, the use of a regular longitude/latitude grid and its extremely fine grid-spacing near the poles will eventually cause a barrier to further operational global resolution upgrades. For this reason, research has already started on the next-generation dynamical core (named GungHo) which we expect to replace ENDGame in the next decade. Being developed in collaboration between Met Office scientists, NERC academics from across the UK and STFC computational scientists from the Hartree Centre, GungHo will be part of a completely new Unified Model that will deliver the step change in scalability required to continue to exploit future generations of computers.



The development of GA6 has benefited from the coordination of effort provided by a seamless model development process. Rather than a large number of scientists and scientific software engineers working solely on upgrading the dynamical core in their system and focusing on their own performance measures, we were able to focus the same amount of effort on upgrading the GA "trunk" configuration and studying a wide basket of metrics and measures. There were several instances of problems and issues identified in one system, that when addressed, improved the performance of another. This meant that the amount of testing that had gone into the configuration as a whole by the time it was implemented was greater than has happened with previous upgrades of a similar size. Whilst Sect. 4 shows that there are still a small number of differences between our "trunk" GA configuration and what has been implemented for global operational NWP, the number of these differences has been reduced and those that remain highlight areas where further improvements are required in either the formulation or our understanding and implementation of the model's parametrisations, which otherwise may not have been exposed.

Over the past two years, GA6/GL6 has been implemented across a wide number of systems and timescales, as illustrated in Table 5. This list is not comprehensive as it does not include implementations and use by collaborating national meteorological

| System | Configuration/system related options | Date implemented/used |
|---|---|---|
| Global NWP suite | N768 GA6.1/GL6.1 deterministic global model | July 2014 |
| | N400 GA6.1/GL6.1 24 member global ensemble | July 2014 |
| Monthly-to-Seasonal forecast system | N216 GA6.0/GL6.0 (as part of coupled GC2) | February 2015 |
| Decadal prediction system | N216 GA6.0/GL6.0 (as part of coupled GC2) | December 2014 |
| Idealised climate change experiments | N96/N216 GA6.0/GL6.0 (as part of coupled HadGEM3-GC2) | Throughout 2014–15 |
| Air quality forecast model | 12 km GA6.0/GL6.0 limited area UK domain | March 2016 |
| | with prognostic chemistry and aerosol fields | |

**Table 5.** A sample of Met Office operational prediction systems that have implemented configurations based on GA6 and the date of their implementation.

centres and academic institutions or non-operational Met Office systems such as our regional reanalysis or our weakly coupled data assimilation/global coupled forecast demonstration system. It also includes our first GA implementation in a limited area modelling system. This reflects the fact that the "Global Atmosphere" configuration is now the recommended science configuration for all UM systems using parametrised convection, including limited area models with grid-spacing $\Delta x \geq 10\,\mathrm{km}$. In contrast, "Regional Atmosphere" configuration development will focus primarily on convection-permitting models with $\Delta x \leq 4\,\mathrm{km}$.

Since the freeze of GA6/GL6, our model development work has focused on further improving physical parametrisations to address known biases in the model and the inclusion of new functionality required for climate simulations contributing to the 6[th] Coupled Model Intercomparison Project (CMIP6, Eyring et al., 2015). This will culminate in the freeze of the Global Atmosphere 7.0 and Global Land 7.0 (GA7.0/GL7.0) configurations, which will be documented in due course. In addition to



being used to further upgrade our operational systems, GA7.0/GL7.0 as part of Global Coupled 3.0 (GC3.0) will form the physical basis of the UK's next Earth System Model (UKESM1).

**Code availability**

*Intellectual property.*

Due to intellectual property right restrictions, we cannot provide either the source code or documentation papers for the UM or JULES. Supplementary material to this paper does include a set of Fortran namelists that define the configurations in the atmosphere/land-only climate simulations at N96 resolution as well as changes that should be made to use the configurations in different systems and at different horizontal resolutions.

*Obtaining the UM.*

The Met Office Unified Model is available for use under licence. A number of research organisations and national meteorological services use the UM in collaboration with the Met Office to undertake basic atmospheric process research, produce forecasts, develop the UM code and build and evaluate Earth system models. For further information on how to apply for a licence see http://www.metoffice.gov.uk/research/collaboration/um-collaboration

*Obtaining JULES.*

JULES is available under licence free of charge. For further information on how to gain permission to use JULES for research purposes see https://jules.jchmr.org/software-and-documentation

**Appendix A:  Breakdown of changes between GA5.0/GL5.0 and GA6.0/GL6.0**

Here, we outline which of the changes discussed in Sect. 3 were introduced in GA5.0/GL5.0 and which were introduced in GA6.0/GL6.0.

**A1    Changes introduced in GA5.0/GL5.0**

– **GA:#10:** Implement the 5A gravity wave drag scheme

– **GA:#18:** Implementation of the ENDGame dynamical core

– **GA:#32:** Connect autoconversion droplet number to aerosol climatologies

– **GA:#43:** Use mixed-phase cloud amount prognostic

– **GA:#49:** A series of safety tests to improve convection



- **GA:#63:** Minor revision to current CLASSIC aerosol dry deposition scheme

- **GA:#65:** Use a consistent droplet number for the first and second indirect effects

- **GA:#70:** Reduce the full radiation timestep to 1 hour

- **GA:#74:** Increase entrainment rate to a multiple of GA3 profile

- **GA:#75:** Revert slow physics to using specific humidity

- **GA:#78:** Consistent use of volume averaging in grid transformations

- **GA:#96:** Update land albedo climatology

- **GL:#8:** Improved treatment of the surface albedo

- **GL:#32:** Increase roughness lengths over sea-ice to GA3.1 values

**A2 Changes introduced in GA6.0/GL6.0**

- **GA:#93:** Address bug in the ENDGame theta source term

- **GA:#94:** Include conserved dry mass in calculating density within the aerosol scheme

- **GA:#106:** Hermite cubic interpolation in the vertical for semi-Lagrangian advection of theta

- **GA:#124:** Tune the non-orographic gravity wave drag scheme

- **GA:#126:** Update to ENDGame dry-mass fixer

*Acknowledgements.* DC, SH, CH, MR, RL, GM, CS, KW and PX were supported by the Joint DECC/Defra Met Office Hadley Centre Climate Programme (GA01101). NK and SW were supported by the National Centre for Atmospheric Science, a Natural Environment Research Council collaborative centre, under contract R8/H12/83/001. The development and assessment of the Global Atmosphere/Land configurations is possible only through the hard work of a large number of people, both within and outside the Met Office, that exceeds the
list of authors.





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
