# Peer review of "The Met Office Unified Model Global Atmosphere 6.0/6.1 and JULES Global Land 6.0/6.1 configurations"

_Geoscientific Model Development, 2016_

## Short Comment (SC1) · 15 Aug 2016

I very much hope that this paper is published. I would suggest that some details about the ancillaries in Table 1 are not quite correct.

1) Are the GA6 soil properties only using HWSD or are other datasets also used? For the United States region, is the State Soil Geographic Database (Miller and White, 1998) used? Are point observations of soil sand, silt and clay fractions (Batjes, 2009) used?

Batjes, N. H. (2009), Harmonized soil profile data for applications at global and continental scales: updates to the WISE database. Soil Use and Management, 25: 124–

127. doi: 10.1111/j.1475-2743.2009.00202.x

Miller, D. and White, R.: A conterminous United States multilayer soil characteristics dataset for regional climate and hydrology modeling, Earth Interactions, 2, 1–26, doi:10.1175/1087-3562(1998)002<0001:ACUSMS>2.3.CO;2, 1998.

2) Is canopy height based on MODIS data as suggested in Table 1 or is it based on IGBP landcover?

3) For the "Urban Canopy" perhaps it would be worth also referencing Best et al (2006) which shows some limitations with the simple scheme. As well as mentioning the MORUSES scheme which is used in the convective scale versions of the Unified Model (Porson et al, 2010).

Best, M. J., C. S. B. Grimmond, and Maria Gabriella Villani. "Evaluation of the urban tile in MOSES using surface energy balance observations." Boundary-Layer Meteorology 118.3 (2006): 503-525.

Porson, A., Clark, P. A., Harman, I. N., Best, M. J., & Belcher, S. E. (2010). Implementation of a new urban energy budget scheme in the MetUM. Part I: Description and idealized simulations. Quarterly Journal of the Royal Meteorological Society, 136(651), 1514-1529.

---

## Author Comment (AC1) · 27 Oct 2016

**1   Reply to specific comments**

Many thanks for your support for this paper and your helpful comments on the details of some of its content. We address each of these in turn below.

**1.1   Soil properties**

*"Are the GA6 soil properties only using HWSD or are other datasets also used?"*

Yes, you are correct that these are really a blend of HWSD and the other datasets you have referenced. The details of this blending is not published, but we have updated table 1 to reflect the source data used.

**1.2 Canopy height**

*"Is canopy height based on MODIS data as suggested in Table 1 or is it based on IGBP landcover?"*

Yes, again, you are correct. The canopy height is currently held in the same file as the leaf area index, which was calculated from MODIS data, but it is actually calculated from IGBP data. Again, we have clarified this in an updated version of table 1.

**1.3 Urban scheme**

*"For the "Urban Canopy" perhaps it would be worth also referencing Best et al (2006) which shows some limitations with the simple scheme. As well as mentioning the MORUSES scheme which is used in the convective scale versions of the Unified Model (Porson et al, 2010)."*

The aim of this paper is not to document the available options within the UM or JULES, but to specifically describe how these are used in our Global Atmosphere and Global Land configurations. To date, the improvement of the urban scheme has focussed on non-GA/GL convection permitting configurations of UM/JULES, so we believe that it will be best to leave the discussion of this issue to the upcoming documentation of those configurations.

**1.4  Updated version of table 1**

To address comments 1 and 2, we propose including the following updated version of table 1 in the final manuscript.

[Figure]

| Ancillary field | Source data | Notes |
|---|---|---|
| Land mask/fraction | System dependent | |
| Mean/sub-grid orography | GLOBE 30″; Hastings et al. (1999) | Fields filtered before use |
| Land usage | IGBP; Global Soil Data Task (2000) | Mapped to 9 tile types |
| Soil properties | HWSD; Nachtergaele et al. (2008) | Three datasets blended via optimal interpolation |
| | STATSGO; Miller and White (1998) | |
| | ISRIC-WISE; Batjes (2009) | |
| Leaf area index | MODIS collection 5 | 4 km data (Samanta et al., 2012) mapped to 5 plant types |
| Plant canopy height | IGBP; Global Soil Data Task (2000) | Derived from land usage and mapped to 5 plant functional types |
| Bare soil albedo | MODIS; Houldcroft et al. (2008) | |
| Snow free surface albedo | GlobAlbedo; Muller et al. (2012) | Spatially complete white sky values |
| TOPMODEL topographic index | Verdin and Jensen (1996) | |
| SST/sea ice | System/experiment dependent | |
| Ozone | SPARC-II; Cionni et al. (2011) | Zonal mean field used[%] |
| Aerosol emissions/fields: | | Only required for prognostic aerosol simulations |
| Main primary emissions | CMIP5; Lamarque et al. (2010) | Includes $SO_2$, DMS, soot, OCFF, biomass burning |
| Volcanic $SO_2$ emissions | Andres and Kasgnoc (1998) | |
| Sulphur-cycle offline oxidants | STOCHEM[*] Derwent et al. (2003) | |
| Ocean DMS concentrations | Kettle et al. (1999) | |
| Biogenic aerosol ancillary | STOCHEM[*]; Derwent et al. (2003) | |
| CLASSIC aerosol climatologies | System/experiment dependent | Used when prognostic fields not available |
| TRIP river paths | 1 data from Oki and Sud (1998) | Adjusted at coastlines to ensure correct outflow |

**Table 1.** Source datasets used to create standard ancillary files used in GA6.0/GL6.0. [*]STOCHEM denotes that these fields are derived from runs of the STOCHEM chemistry model. [%]This is expanded to a "zonally symetric" 3D field in limited area simulations on a rotated pole grid.

---

## Referee Comment (RC1) · Anonymous Referee #1 · 1 Nov 2016

The manuscript describes the latest operational configuration of the UM. As this is a full operational model coupled to a land model, there is a tremendous amount of topics to be considered in this manuscript. It is praiseworthy how the authors manage to keep an overview over the various topics and point the readers to in-depth literature where necessary. Therefore, the paper is interesting and informative to read in all its sections.

From the scientific viewpoint I found the positive impact of a less inherently damping dynamical core interesting and well documented. The tuning of for instance the convection scheme is tracable documented.

Encouraging are the improved results, the better scaling on the multi-processor machine, and the obviously well-organized code tracking system.

---

## Referee Comment (RC2) · Anonymous Referee #2 · 27 Dec 2016

I concur with the earlier comments of referee #1.
* * *

---

## Author Comment (AC3) · 5 Jan 2017

A joint response to both referees and the additional comment are included in Author Comment **AC2**

---

## Author Response (AR1)

**Author's response to reviews/comments on "The Met Office Unified Model Global Atmosphere 6.0/6.1 and JULES Global Land 6.0/6.1 configurations" by D. N. Walters et al.**

D. N. Walters et al.

*Correspondence to:* D. N. Walters
(david.walters@metoffice.gov.uk)

**1 Comments from referees**

Both referees have made positive comments about the discussions paper and have not suggested or requested any changes to this ahead of publication.

**2 Author's response**

5 We thank both referees for their reviews and for their support for the publication of this paper. Given how widely the Global Atmosphere/Land configurations are used, we believe it to be an important part of our development/implementation process to produce a peer-reviewed paper documenting the configuration as a whole, as well as highlighting the changes made since the previous configuration and the impacts these have on model performance.

**3 Comments from other contributors to the discussion**

10 In addition to the reviews, there were some specific questions from Imtiaz Dharssi at the Bureau of Meteorology in Australia about our description of some of the land surface ancillary data. Imtiaz was involved in the development and implementation of these ancillaries during his previous employment at the Met Office and is therefore particularly well placed to comment on the details of their description.

His specific comments were:

1. Are the GA6 soil properties only using HWSD or are other datasets also used? For the United States region, is the State Soil Geographic Database (Miller and White,1998) used? Are point observations of soil sand, silt and clay fractions (Batjes, 2009) used?

2. Is canopy height based on MODIS data as suggested in Table 1 or is it based on IGBP landcover?

3. For the "Urban Canopy" perhaps it would be worth also referencing Best et al (2006) which shows some limitations with the simple scheme. As well as mentioning the MORUSES scheme which is used in the convective scale versions of the Unified Model (Porson et al, 2010).

Imtiaz's full comments are available in discussion comment **SC1**

**4 Author's response**

Imtiaz's comments were most welcome and again highlight the benefit of an open discussion on these papers. They have allowed us to improve the accuracy of our documentation, which is of benefit to us as well as to the users of our configurations.

A full reply to Imtiaz's comments area available in discussion comment **AC1**, but we include the main reply below for completeness:

1. *Soil properties*: Yes, you are correct that these are really a blend of HWSD and the other datasets you have referenced. The details of this blending is not published, but we have updated table 1 to reflect the source data used.

2. *Canopy height*: Yes, again, you are correct. The canopy height is currently held in the same file as the leaf area index, which was calculated from MODIS data, but it is actually calculated from IGBP data. Again, we have clarified this in an updated version of table 1.

3. *Urban scheme*: The aim of this paper is not to document the available options within the UM or JULES, but to specifically describe how these are used in our Global Atmosphere and Global Land configurations. To date, the improvement of the urban scheme has focussed on non-GA/GL convection permitting configurations of UM/JULES, so we believe that it will be best to leave the discussion of this issue to the upcoming documentation of those configurations.

**5 Author's changes to manuscript**

Following Imtiaz's suggestions in discussion comment **SC1**, we have updated table 1 as discussed above to more accurately cite the source data used for certain land surface ancillaries.

In addition to this, we have also made the following changes as highlighted in the attached latexdiff created pdf:

1. In Sect. 2.11, we have corrected an error in the description of the "inland water canopy". Whilst some configurations of JULES assign the lake canopy with a heat capacity of $4.18 \times 10^6 \, \mathrm{J \, K^{-1} \, m^{-2}}$ (which is the equivalent of $\approx 1\mathrm{m}$ depth of water), the GL configuration uses $2.11 \times 10^7 \, \mathrm{J \, K^{-1} \, m^{-2}}$ (i.e. $\approx 5\mathrm{m}$ depth), which is believed to be more representative of lakes globally.

2. We have improved the consistency of the labelling in sub-sub-sections of Sect. 3.

3. We have updated a URL cited in the "Code availability" section.

[revised manuscript text omitted]

small, but almost always beneficial and is achieved without affecting the variability of the forecast as measured by the standard deviations (not shown). Similar results have been found in full data assimilation trials run over multiple periods and with multiple baseline configurations.

The CAPE timescale of 1 h used in GA6.0 was chosen as a compromise between two extremes. Operationally, however, it has been hard to justify the small but consistent reduction in predictability associated with increasing the CAPE timescale from the previously operational value of 30 min used in GA3.1. For this reason, the GA6.1 configuration used for operational global NWP continues to use this shorter CAPE timescale. Our belief is that the lack of a single parameter value suitable for all purposes exposes a weakness in the current parametrisation and suggests that an alternative approach is required, such as a dynamically diagnosed CAPE timescale or an alternative convective closure.

**4.2 Land surface and hydrology: Global Land 6.1**

**4.2.1 Aggregated surface tile**

In addition to the CAPE timescale, another long-standing difference between operational global NWP and other operational configurations of the UM is that the former (including GL3.1) has always performed its land surface calculations over a single land surface tile with the aggregated properties of the 9 individual surface types rather than performing these in parallel and aggregating the resulting fluxes. Initial investigations have shown that this is due to the Bowen ratio (i.e. the ratio of sensible to latent heating at the land surface) being higher in the 9 tile model, leading to large near-surface warm biases and near-surface low pressure biases in some regions during local summer.

It is not yet clear whether the "improved" performance of the aggregated tile is due to a deficiency in the 9 tile approach (possibly due to errors in the specification of surface parameters) or due to some aspect of the global NWP system having been developed to perform well with a 1 tile model (e.g. the details of the land surface data assimilation). In the absence of having made progress in understanding this issue, therefore, GL6.1 continues to use the aggregated tile approach that was used operationally with GL3.1. Because the aggregated tile approach is incompatible with holding snow on the vegetation canopy and with the use of the "inland water canopy" for modelling lakes, these schemes are also dropped from GL6.1. Finally, it is impossible to sensibly aggregate the thermal and momentum roughness lengths (respectively labelled $z_{0h}$ and $z_{0m}$) using the range of values of $z_{0h}/z_{0m}$ from Table 3 of Walters et al. (2014), so in GL6.1 the value of $z_{0h}/z_{0m}$ for broadleaf and needle-leaved trees is reduced from the GL6.0 value of 1.65 to the GL3.0 value of 0.1.

**4.2.2 Thermal conductivity of sea ice**

Rae et al. (2015) describes the development of the Global Sea Ice 6.0 (GSI6) configuration of the Los Alamos CICE sea ice model (Hunke and Lipscombe, 2010), which was developed in parallel to GA6.0/GL6.0 for use in coupled simulations as part of the Global Coupled model 2.0 (GC2) configuration (Williams et al., 2015). For consistency between the Global Land configuration in coupled and uncoupled simulations, where changes to GSI6.0 included changes to the JULES land surface model, we have included these same changes in our GA/GL simulations.

For one set of parameters, namely the thermal conductivity of sea ice and snow on top of sea ice (labelled $\kappa_{ice}$ and $\kappa_{snow}$ respectively) we omitted to make these changes in pre-operational NWP tests of GA6.1/GL6.1. Rather than fixing this issue, which would have required an additional round of trialling and a delay to operational implementation, we decided to include this change in the definition of GL6.1. The values of these parameters are shown in Table 4. As the presence of this difference was accidental, this will be removed in the next Global Land release. With prescribed sea ice fractions and thicknesses, the impact of these differences on an uncoupled GA/GL simulation are small, but non-zero. This is because the sea ice in these simulations is specified with a fixed temperature at ice base, such that the sea ice surface temperature is dependent on its thermal conductivity. As shown in Fig. 8, in the winter hemisphere, where the near-surface air temperature is much colder than the freezing point of sea water, the reduced thermal conductivity in GL6.1 leads to a warmer surface temperature over sea ice.

| Parameter | GL6.0 (& GSI6.0) | GL6.1 |
|-----------|------------------|-------|
| $\kappa_{\text{ice}}$ | $2.63\,\text{W}\,\text{m}^{-1}\,\text{K}^{-1}$ | $2.09\,\text{W}\,\text{m}^{-1}\,\text{K}^{-1}$ |
| $\kappa_{\text{snow}}$ | $0.50\,\text{W}\,\text{m}^{-1}\,\text{K}^{-1}$ | $0.31\,\text{W}\,\text{m}^{-1}\,\text{
[revised manuscript text omitted]